# Modeling the Effect of Scrap on the Electrical Energy Consumption of an Electric Arc Furnace

**Leo S. Carlsson \*, Peter B. Samuelsson** **and Pär G. Jönsson**

Royal Institute of Technology , Brinellvägen 23, 114 28 Stockholm, Sweden; petersam@kth.se (P.B.S.); parj@kth.se (P.G.J.)

\* Correspondence: leoc@kth.se

**Abstract:** The melting time of scrap is a factor that affects the Electrical Energy (EE) consumption of the Electric Arc Furnace (EAF) process. The EE consumption itself stands for most of the total energy consumption during the process. Three distinct representations of scrap, based partly on the apparent density and shape of scrap, were created to investigate the effect of scrap on the accuracy of a statistical model predicting the EE consumption of an EAF. Shapley Additive Explanations (SHAP) was used as a tool to investigate the effects by each scrap category on each prediction of a selected model. The scrap representation based on the shape of scrap consistently resulted in the best performing models while all models using any of the scrap representations performed better than the ones without any scrap representation. These results were consistent for all four distinct and separately used cleaning strategies on the data set governing the models. In addition, some of the main scrap categories contributed to the model prediction of EE in accordance with the expectations and experience of the plant engineers. The results provide significant evidence that a well-chosen scrap categorization is important to improve a statistical model predicting the EE and that experience on the specific EAF under study is essential to evaluate the practical usefulness of the model.

**Keywords:** electrical energy consumption; Electric Arc Furnace; scrap melting; statistical modeling

## 1. Introduction

Electrical Energy (EE) can account for between 40–66% of the total energy usage during the Electric Arc Furnace (EAF) process, which is a number that highlights the importance of further improvements in modeling of the EE consumption [1]. The energy losses, which partly governs the EAF process energy dynamics, are mostly related to off-gases, slag, dust, furnace cooling, electrical and radiative losses. Most of these energy losses are closely linked to the total process time of any given heat. The process time itself is influenced by numerous impositions. One of these is the melting time of the charged raw materials, which in the scope of this study is primarily steel scrap.

Many articles have studied and proposed statistical models predicting the EE consumption of the EAF [1,2]. However, only a handful of studies have used scrap types as input variables to a statistical model predicting the EE of an EAF [2–7]. Neither of these studies analyze the contributions of each scrap type on specific predictions by the statistical model. Verifying the effects of the input variables on the complete prediction space is paramount to evaluate the practical usefulness of any statistical model let alone to make the users, i.e., process engineers, trust the model. Only when these two conditions are met can the model possibly be used to solve practical problems. An example of a practical problem where an EE prediction model can be used is when determining the EE requirement for the EAF in a Demand Side Management (DSM) system, which optimizes the processes in a steel plant with respect to the available power in the transmission line [8].

Shapley Additive Explanations (SHAP) is a recent development in the field of interpretable machine learning [9], and has previously been used to analyze a statistical model predicting the EE of an EAF producing stainless steel [10]. Tap-to-Tap time (TTT), delays, and total charged weight were found to be the three most influencing variables. TTT and total charged weight were correctly interpreted by the model with respect to what is known from process metallurgical experience. The delay variable was incorrectly interpreted by the model, which was concluded to be due to the high correlation between delays with TTT.

The aim of the current study is to investigate the effect of scrap types on a statistical model predicting the EE of an EAF. Although verifying the effects of other input variables on the model output is important, it is not the main focus of this article. For such an analysis we refer to previous studies [2,10]. To investigate the effects of scrap, three distinct representations of scrap types based on the plant scrap codes, scrap physical shape, and scrap apparent density, respectively, will be used in the models. The reasons are two-fold. First, to provide the steel plant engineers with an intuitive and simple method to categorize scrap for modeling purposes. Second, to find the optimal scrap representation for the prediction problem with respect to the accuracy and precision of the statistical model.

In addition to its scrap-oriented focus, this study further builds on the modeling methodology presented in two previous studies [2,10]. Four different data cleaning strategies will be used to investigate the effect of data cleaning on the accuracy of the statistical model and an additional non-linear statistical model framework will be employed; Random Forests (RF). Furthermore, this study uses data from an EAF producing steel for tubes, rods, and ball-bearing rings and not from an EAF producing stainless steel, thus broadening the application of the modeling methodology.

The results demonstrated that the three subsets of input variables provided by the scrap representations all increase the performance of the models. Using SHAP, it was found that heavy scrap, i.e., scrap with low surface-area-to-volume ratio, contributed to an increased EE consumption while steel sheets, a scrap type with high surface-area-to-volume ratio, contributed to a decreased EE consumption. These findings were confirmed by the steel plant engineers to agree well with previous experiences using these scrap types as raw material.

## 2. Background

### 2.1. Melting of Steel Scrap in Liquid Steel

#### 2.1.1. Driving Forces

The driving forces in scrap melting are present in any process in which scrap melting occurs. However, the driving factors vary significantly between the EAF and Basic Oxygen Furnace (BOF). This section highlights the driving factors in scrap melting with respect to the EAF in the steel plant of study. The goal is not to create an extensive review over the various research topics governing the melting of scrap, but rather to motivate the scrap representations used in the experimental part of this study. A comprehensive review in the field was compiled by Freidrich [11]. Prominent later developments in the field of scrap melting have been summarized in a recent review [12].

The melting of solid scrap in liquid steel is dominated by several factors. These are temperature gradients between solid scrap and liquid steel, concentration gradients between solid scrap and liquid steel, the freezing effect, the rate of stirring of the steel melt. These phenomena are explained further below.

**Temperature gradients** between the solid scrap and the steel melt is one of the most important driving factors in the melting of steel scrap. The melting rate, in m/s, can be determined by the following equation:

$$\frac{dx}{dt} = h \cdot \frac{T_{HM} - T_{liq}}{\rho_{scr} \cdot (H_s + (T_{HM} - T_{liq}) \cdot c_p)} \tag{1}$$

where $T_{HM}$ is the temperature of the molten steel and $T_{liq}$ is the scrap melting temperature. Furthermore, $H_s$ is the heat of melting of scrap, $c_p$ is the specific heat of scrap, $\rho_{scr}$ is the density of the scrap metal, and $h$ is the heat transfer coefficient in the interface of the molten steel and scrap [12]. The higher the temperature gradient, $T_{HM} - T_{liq}$, the faster the steel scrap will melt.

**Alloying element gradients** also contribute to the melting of scrap in a process known as dissolution. In this case, alloying elements migrate to the solid-liquid metal interface. The most dominant alloying element in this process is carbon. However, in the BOF, the carbon concentration difference can exceed 4 wt-% while in the EAF the carbon concentration difference seldom exceeds 1 wt-%.

Assuming that the dissolution rate by carbon can be determined by the shortest length of the scrap, the following equation can be used:

$$\frac{dx}{dt} = \frac{\beta \cdot (C_l - C_i)}{\rho_s(C_l - C_0)} \tag{2}$$

where $\beta$ is the mass transfer coefficient of carbon, $C_l$ is the carbon content in the liquid steel, $C_0$ is the initial carbon content in the steel scrap, and $C_i$ is the carbon content in the solid-liquid interface [13].

A similar equation can be determined for silicon, which can also be an alloying element to account for should the difference in silicon concentration between steel scrap and the molten steel be large.

The **freezing effect** occurs in the solid-liquid interface when the scrap first comes in contact with the liquid steel. A solidified shell is formed due to the large temperature difference between the two. This means that the volume of the scrap increases initially. The solidified shell is proportional to the surface area of the scrap that is submerged in the hot metal or molten steel. Hence, the reduction in the steel scrap size does not occur instantly, rather it decreases after the solidified shell has melted.

The **stirring** velocity is the velocity of the melt in the boundary layer between the melt and the scrap surface area. Numerous studies have related the stirring velocity to the mass transfer coefficient on scrap in liquid steel. However, there exists a wide range of reported mass transfer coefficient values for scrap in liquid steel under forced convection [12]. Nevertheless, a commonly deduced relationship between the mass transfer coefficient and the stirring velocity may be written as follows:

$$h_{scr} = c \cdot u^p \tag{3}$$

where $h_{scr}$ is the mass transfer coefficient under forced convection, $c$ and $p$ are constants that are determined experimentally. $u$ is the average stirring power, which is related to the average stirring velocity due to the physical relationship between energy, momentum and velocity. The stirring power is governed by, for example, oxygen blowing and carbon boil.

Furthermore, the effect of stirring on the melting rate of scrap in the EAF is low compared to the BOF since the stirring is more intense in the latter, i.e., higher stirring velocity governed by the stirring power per unit volume. Furthermore, one should not expect the stirring to be very intense in the EAF since the liquid steel depth is low and the solid-to-liquid ratio is high prior to the final stages of the process, i.e., superheating, which hampers the flow velocity of the liquid steel. The device that primarily facilitates stirring in the EAF is oxygen lancing, but other devices such as porous plugs and induction stirring enhance the stirring.

### 2.1.2. Scrap Surface-Area-to-Volume Ratio

It is evident that the aforementioned factors are dependent on the surface-area-to-volume ratio of the scrap pieces. On the one hand, the effect from temperature and alloying element gradients influence on the complete surface area exposed to the steel melt. On the other hand, the mass of the steel scrap piece determines the melting time since more mass needs to be heated (Equation (1)) and more mass of the alloying elements have to be transported (Equation (2)). The mass is proportional to the volume of the scrap piece. Thus, the surface-area-to-volume ratio can be expressed either as a

function of the surface area and volume or as a function of the surface area, apparent density of scrap, and mass of the scrap piece:

$$R_{SV} = \frac{A}{V} = \frac{A \cdot \rho_s}{m} \tag{4}$$

Hence, to facilitate lower melting times one should use scrap that has a high surface-area-to-volume ratio. The surface-area-to-volume ratios of some elementary geometrical shapes are presented below to illuminate the effects of the surface-area-to-volume ratio in scrap melting. However, real scrap pieces are often of more complex geometric shapes.

**Sphere:** $\frac{A}{V} = \frac{4\pi r^2}{\frac{4\pi r^3}{3}} = \frac{3}{r}$

**Cylinder:** $\frac{A}{V} = \frac{2\pi rl + 2\pi r^2}{\pi r^2 l} = 2(\frac{1}{r} + \frac{1}{l})$

**Cube:** $\frac{A}{V} = \frac{6l^2}{l^3} = \frac{6}{l}$

**Square plate:** $\frac{A}{V} = \frac{2l^2 + 4lt}{l^2 t} = 2(\frac{1}{t} + \frac{2}{l})$

The thickness is defined as the thinnest dimension of the scrap piece. For the cylinder, one ought to keep $r << l$. For the square plate, one ought to keep $t << l$. The cube and sphere are equidistant from the center of the scrap piece, which means that one should keep the length and radius as small as possible, respectively.

### 2.1.3. The Steel Plant of Study

The dissolution of steel scrap in molten steel due to carbon content gradients between the steel scrap and molten metal are not significant in the EAF of study. The steel plant does not produce high Si steels nor does the carbon content vary significantly between the hot heel and the charged scrap. The temperature gradients between the steel scrap and molten steel will be similar for all heats which have the 5–10-ton hot heel remaining in the furnace at the start of the heat. However, some heats will be produced without this initial 5–10 ton of hot heel. This will affect the melting through temperature gradients mainly for the scrap charged by the first basket. The main source of stirring in the EAF of study is by oxygen lancing. The stirring is mainly facilitated by CO from carbon boil.

### 2.2. The Electric Arc Furnace

#### 2.2.1. Process

The steel plant of study uses an EAF with a nominal charging capacity of 110 ton and a transformer system of 80 MVA. The steel products produced are rods, tubes, and ball-bearing rings. The EAF does not use a pre-heater but uses a hot heel, which is molten steel left over from each previous heat. The amount of hot heel is 5–10 ton. During operation oxyfuel burners are used to remove the cold spots between the furnace electrodes. The burners can also function as oxygen lances to inject oxygen to the molten metal.

The process begins with a default mode where 5–10 ton of hot heel and a full basket of scrap are present. The scrap basket is layered with different scrap types according to a pre-specified recipe. When the basket of scrap has been charged into the furnace, the lowermost scrap mixes with the hot heel and creates a mixture of scrap and partially molten steel. The melting phase starts when the transformer is powered on and the electrode arcs are bored down into the upper layer of the scrap. This process proceeds until enough space is available for the second basket of scrap to be charged. In this instance, the amount of molten steel has increased but partially molten steel scrap is still present in the steel bath. Furthermore, heaps of scrap are also present around the electrode arcs. The scrap from the second basket gets piled on top of these heaps of scrap. The second melting phase starts in a

similar manner as the first melting phase. When a clear visible molten steel bath is present, the refining phase starts. Oxygen lancing and injected fine carbon facilitate a foaming slag which increases the energy yield from the arcs to the steel bath. Some partially molten steel scraps are present in molten bath during the refining phase. These are usually large and bulky pieces of scrap that require longer exposure time to the molten steel to completely melt. Oxygen lancing also provides stirring of the steel bath, which increases the melting rate of the remaining steel scraps. The contribution of the stirring is expected to be minor compared to stirring using, for example, an electromagnetic field. Finally, the steel is tapped into a ladle and the steel is further treated in downstream processes.

The described EAF process is illustrated in Figure 1.

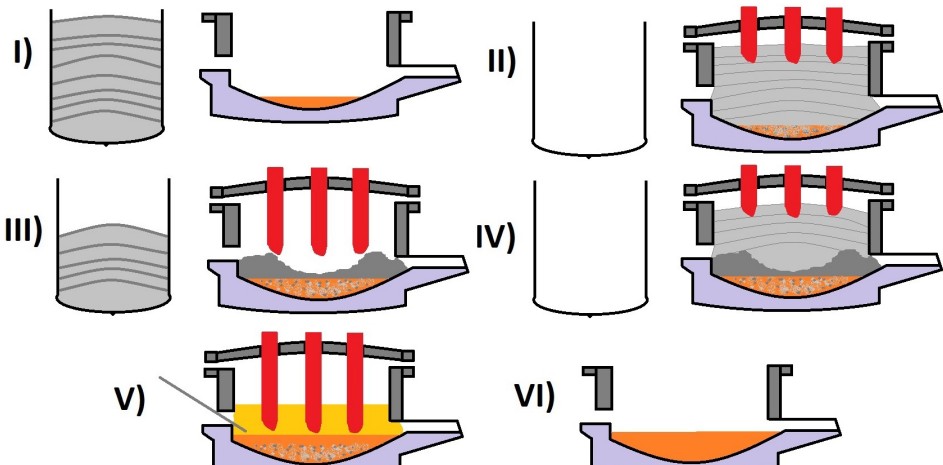

**Figure 1.** The Electric Arc Furnace (EAF) process in the steel plant considered in the current article. It shows the process divided into six parts. (**I**) The first scrap basket is fully charged with up to 10 layers of scrap. The furnace shell contains hot heel. (**II**) The scrap from the first basket has been charged into the furnace whereupon the lowermost scrap layers intermix with the hot heel. The furnace arcs are bored down into the scrap and the transformer is powered on. (**III**) After the first melting phase the furnace contains a steel scrap-melt mixture and heaps of partially solid scrap. The second scrap basket is charged in as much to fully fill the furnace in the next step. (**IV**) Charging of the second basket. (**V**) Refining of the molten steel where injected carbon and oxygen lancing facilitate an even layer of foaming slag. (**VI**) The heat is ready for tapping.

As a last step, any preparations to the furnace are made before the next heat. This can, for example, be replacement of electrodes or fettling of the furnace refractories. The steel plant of study also replaces the hot heel periodically to allow for inspection of the furnace bottom refractories. This means that some heats do not have a hot heel present at the start of the heat.

### 2.2.2. Charging of Scrap Baskets

Several aspects that affect the scrap charging either directly or indirectly must be highlighted. The various steel grades have pre-specified recipes, i.e., charge types, which are used to determine how much of each scrap type that will be charged into the baskets. This, in its turn, is determined by the selection of primary scrap and alternative scrap. Primary scrap is always used if it is available in the scrap yard. Alternative scrap is used if the primary scrap is not available. Each primary scrap type has a matching alternative scrap type. Hence, the recipe is a representation of an ideal charged basket, but not necessarily representative of the actual charged basket.

Available scrap types on the market also affect what charging strategies are possible. The available scrap types in the market set the limit to what performance can be expected with regards to the EE consumption. If a major scrap type that requires less EE to melt is not available, then the EE of the next heats could be expected to increase, if a scrap type with higher apparent density is used instead. Since the scrap turn-over rate can be as low as one week for several scrap types, varying available

scrap types are thus to be expected. Customer demand also influences the scrap requirements since scrap with higher amount of tramp elements cannot be used for steel types with high performance requirements. The market factors and customer demand variations will not be accounted for in the numerical experiments. However, it is important to keep these factors in mind since they indirectly influence the EE consumption.

Some charging strategies are subsequently described. The first basket is usually filled with scrap to 75–100% of the total volume of the basket. The total volume of a scrap basket is 65 m$^3$. The second scrap basket is filled to reach the pre-specified amount of molten steel, which is approximately 100 tonnes. A third scrap basket is used if the amount of required scrap was not satisfied by the first two baskets.

Each scrap basket can be made up of a total of 10 layers of different scrap types. The scrap types in each layer are pre-specified by the scrap recipe. Combined, the layers will consist of a blend of various scrap types. Bulky scrap with high apparent density ($1.4 \geq$ tons/m$^3$) such as internal casting residuals are put in the bottom layers of the basket. This ensures that the bulkier scrap gets a longer exposure to the steel melt and thus reduces the risk of solid scrap pieces in the latter stages of the EAF process. Consequently, scrap with lower apparent density (0.4–1.0 tons/m$^3$) such as sheet and turnings are put in the intermediate and upper layers. These scrap types do not require as long exposure to the steel melt compared to other scrap types. Furthermore, the main source of heat for melting these scrap types, most likely, comes from the radiative heat transfer from the electrode arcs.

Some effect of compaction of scrap types with lower apparent densities is expected due to the weight imposed by the upper layers of scrap. Hence, the apparent density of these scrap types will be higher in a fully charged basket compared to when they are solely present in the basket or in the scrap yard. The apparent density of the scrap in the charged furnace can be assumed to have a reasonable agreement with the apparent density of the scrap in the scrap bucket post-charging.

### 2.2.3. Parameters Governing the EE Consumption

Several previous articles have presented estimates of the energy sources and energy sinks during the EAF process [14–18]. These reported values have been compiled in Table 1 and presents a guidance regarding which input variables that need to be considered when predicting the EE consumption. The reason behind the large percentage differences in energy sources and energy sinks is because one of the studies compiled information from 16 EAF [14], some of which used up to 90% Direct Reduced Iron (DRI) and 10% scrap as raw materials. Due to the gangue content of the DRI, mainly $SiO_2$, more slagformers must be added which increases the energy consumption. Furthermore, DRI contains some remaining iron oxides that will be reduced by carbon in the melt. This also requires additional energy. In addition, the amount of injected oxygen per ton charged raw material ranged from 5 m$^3$/t to 40 m$^3$/t, which contributes to the large difference in oxidation of alloying elements [14]. The other articles provided data on one EAF each, three of which used 100% scrap [15,17,18] and one that used an unspecified mix of scrap and DRI [16].

A further discussion about the choice of input variables related to the energy balance equation, but not related to scrap charging, has been published previously [1].

**Table 1.** Synthesized values of energy sources and energy sinks reported in [14–18].

|  | Energy Factor | % of Total Energy Sources or Energy Sinks |
|---|---|---|
| **In** | Electric | 40–66% |
|  | Oxidation of alloying elements | 20–50% |
|  | Burner fuel | 2–11% |
| **Out** | Liquid steel | 45–60% |
|  | Slag and dust | 4–10% |
|  | Off-gas | 11–35% |
|  | Cooling | 8–29% |
|  | Radiation and electrical losses | 2–6% |

In addition, the EE consumption is partly governed by the selected scrap mix in the steel plant. This has been schematically illustrated in Figure 2. The available scrap in the scrap yard sets the limit to the charged scrap mixture in each basket. The charging strategy, comprising of both the scrap types and the layering of the basket, determines the exposed surface area of each scrap piece to the hot heel and the furnace arcs during the process. The burners and oxygen lancing facilitate the melting of the scrap. These factors in combination determine the aggregated melting time of the baskets combined. An increased melting time contributes to a longer TTT, which increases the total heat losses during the process. The increased heat loss has to be counteracted by increased amount of energy sources, which is a combination of EE consumption, burner fuel, and oxidation by alloying elements in the scrap (Table 1).

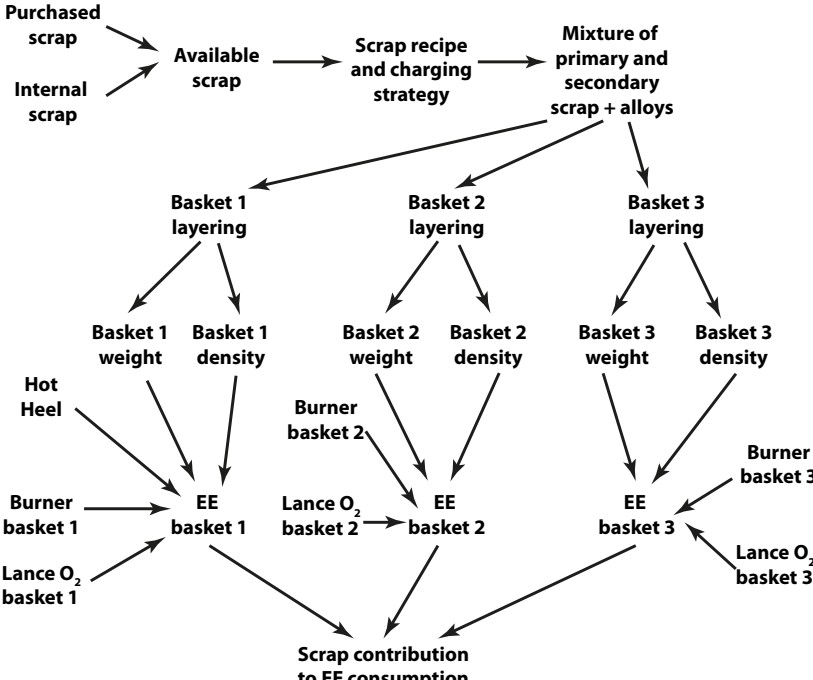

**Figure 2.** The relationship between the available scrap and the scrap contribution to the Electrical Energy (EE) consumption.

### 2.2.4. Non-Linearity

The non-linearity of the EAF process with respect to EE is caused by numerous factors. One of these factors is related to the various sub-processes of the EAF as well as the resulting total process time. For example, the heat loss through radiation and cooling losses are more prevalent during the refining stage, when the steel is molten, compared to the first charging and melting phases. Furthermore, the radiative heat transfer is proportional to $T^4$, which adds to the non-linearity. By varying the refining times and melting times, the resulting contribution to the EE consumption can vary significantly between heats.

Delays imposed by up- and downstream processes as well as by the process itself add to the non-linearity of the process with respect to the EE consumption. The reason is because the process time is no longer a result of the total charged scrap weight or produced steel grade. Furthermore, the delays are very hard to predict and should therefore be considered to be an external non-linear factor imposed on the EAF at all times. The effects of delays have been extensively discussed in [1].

The charged scrap types also add to the non-linearity of the process since combining varying amounts of each scrap type will affect the melting behavior. For example, large and bulky pieces of steel scrap must be exposed to the steel melt for longer times than thinner pieces of steel scrap.

Larger fractions of heavy scrap increase the melting time, which in turn increases the EE consumption. The general strategy is to charge bulkier scrap types in the lower layers of the scrap bucket. This is regardless of the amount of bulky scrap in the heat. Failing to expose bulkier scrap to the hot heel and molten steel means that partially molten pieces of scrap will remain during the latter stages of the refining process. Hence, the melting time becomes the factor that determines the TTT of the heat and hence also the EE consumption.

*2.3. Statistical Modeling*

2.3.1. Inherent Traits

There is one key difference that separates statistical models from physico-chemical models and that is the connection between the input and output values. Physico-chemical models present their prediction, i.e., output value, based on pre-determined equations that use the input values. These equations are related to established physical and chemical laws. Statistical models, on the other hand, interpret the values of the input variables in the context of previously observed input values and output values. Hence, the output of a statistical model is purely based on probability and does not necessarily adhere to established physical and chemical laws. The connection between the two is purely dependent on the data that is used to adapt the statistical model to the prediction problem of interest. This leads to three distinct traits unique to statistical models. These are data quality, data variability, and correlation.

Data quality is related to how close the registered value is to the true value that is intended to be measured. Uncertainties are imposed and the performance of a statistical model is reduced if the data quality is low. There are numerous sources that can affect the data quality. Two examples are the manual logging of data, which is prone to human error, and the definition of a variable in the logging system, which may differ from what is measured in reality. One common data quality issue in the steel industry is the precision of measurement equipment. For example, scales weighting raw materials can have a precision of $\pm 100$ kg and temperature sensors can have a precision of $\pm 5\,^{\circ}$C. In this study, data quality refers to the extent to which the data is affected by the aforementioned examples.

Data variability is a requirement because variations of the values in previously observed data are what the statistical model learn. An input variable that is constant is useless to a statistical model. A constant variable in a physico-chemical model is not useless. A straightforward example is the latent heat of melting of steel, which is an important component of a physico-chemical model predicting the temperature of steel in, for example, the EAF.

Correlation is a metric that indicates the relation between random variables. Strongly correlated input variables are similar and therefore redundant as input variables to a statistical model. In this case, only one of the input variables should be selected. Weakly correlated variables, on the other hand, may be redundant. The degree of redundancy for weakly correlated variables depends on the intra-correlation between the input variable and the other input variables. For example, scrap type A may be redundant if the total scrap weight and scrap types B and C are included in a potential model. The sum of scrap type B and C implies that scrap type A must be the difference between the total scrap weight and scrap types B and C. In this case, adding scrap type A as input variable would likely not increase the accuracy of the statistical model since there is no new information gained from this input variable. It is important to note that correlation *does not* imply causation. However, statistical models lack the ability to distinguish between the two. Even though a correlation can shed light on areas where causation may exist, it is the task of the practitioner with domain-expertise to separate the causative relations from the non-causative relations. This stresses the importance of possessing knowledge about the domain in which the statistical model is used.

### 2.3.2. The Abstract Case

Supervised statistical models will be used in this paper which means that each row of input data, i.e., variables, has a corresponding output data point. This framework can be explained by the following steps:

1. Select statistical model framework. The available hyper-parameters are unique to the statistical model framework and are selected by the modeler.
2. Train the model using a set of matching input and output data. Continue the training phase until the accuracy of the model converges.
3. Test the model with a set of previously unseen data.
4. Evaluate the models practical usefulness using the accuracy on the test data.
5. If the accuracy is good enough, deploy the model in a production environment.

In a practical context, the model hyper-parameters (step 1) are chosen based on a comprehensive hyper-parameter search, also known as grid-search. During the parameter search, several models are trained for each combination of hyper-parameters. The combination with the highest and most stable accuracy is the most optimal hyper-parameter selection.

Non-linear supervised statistical models should be used when predicting the EE consumption of an EAF. This is because some important input variables governing the EAF process are non-linearly related to the EE consumption. The statistical model framework should always be chosen based on the nature of the prediction problem.

Although non-linear statistical models are excellent at learning complex relations between variables, these types of models are susceptible to overfitting. Overfitting means that the statistical model has adapted itself too well to a particular set of data, i.e., the training data, in such a way that it cannot predict well on future data. Combating this phenomenon is important since the relations between the variables are expected to change from the training data to test data. This is the natural course of any steel plant process. The strategies to reduce overfitting will be explained further in Section 3.4.1 where the specific model frameworks used in the numerical experiments are presented.

### 2.3.3. Previous Studies

Statistical models have previously been used as a tool to predict the EE consumption of the EAF. A comprehensive review of the subject has recently been published [1]. However, only four of the previous studies have used some representation of scrap types as part of the input variables in the models and as part of the model analysis [3–6].

The first study used the weight of shredded scrap as the only scrap type variable for a Multivariate Linear Regression (MLR) model [6]. The coefficient for this variable is negative, which indicates that less EE is needed than what is normally required when more shredded scrap is added. The model was then used on data from 5 different EAF, all of which used various amounts of other scrap types which were not taken into consideration by the model.

The second study used response graphs to investigate the total EE prediction response by each scrap type in the first and second baskets [4]. However, a response graph only displays the total EE prediction when varying one single input variable and does not reveal the specific contribution by each input variable.

The third study used Partial Least Squares (PLS) regression to model the EE consumption of two different EAF [3], one of which is the steel plant governing the data in the current study. However, the significance of each scrap type representation was only given by an ad-hoc subjective measure, as indicated by the descriptive words *low* and *high*.

The last study used the statistical modeling frameworks ML, RF, and Artificial Neural Networks (ANN) to predict the EE consumption [5]. The effect of the scrap types on the EE was only reported for the MLR model since the model coefficients reveal the impact of each scrap type on the EE consumption. The values of the coefficients were then compared with experience-based values.

However, some assumptions were made regarding the comparison since the MLR model used kWh/t charged scrap and the experience-based values were reported in kWh/t tapped steel.

A recent study, published after the review, used Kolmogorov–Smirnov (KS) tests and correlation metrics to highlight the change of all input variables between the training and test data [2]. In addition, permutation feature importance was used to investigate the importance of each input variable to the model prediction for both data sets. In combination, the KS tests and permutation feature importance produced evidence that some of the input variables had the main influence in the performance reduction of the model. However, the focus was never to investigate the specific effects of the scrap types on the EE consumption of the selected models.

## 3. Method

### 3.1. Representing Scrap Types

The scrap types will be represented in three distinct ways. The first representation will use the scrap codes from the steel plant of study, i.e., scrap type. The second representation is based on a visual categorization of each scrap code with the aim to provide an intuitive categorization for the steel plant engineers as well as to minimize the number of distinct scrap categories. The third representation is based on the estimated apparent density of each scrap code. The estimated apparent densities are both from established technical specifications as well as from estimations conducted by the plant engineers. The relations between the three scrap representations can be seen in Figure 3. Henceforth, the term *scrap representation* will refer to either of the three distinct scrap representations and the term *scrap category* will be used to specify a specific category in either of the visual or apparent density categorizations. The term *scrap type* will be used to refer to a specific scrap type as defined by the steel plant coding system.

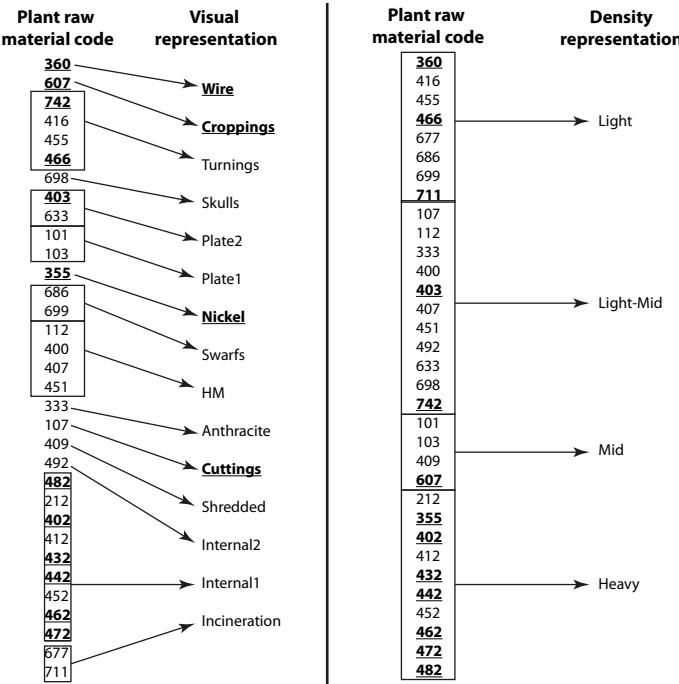

**Figure 3.** The relationship between the steel plant scrap types and the two scrap representations based on visual and estimated apparent density properties, respectively. The bold underlined scrap types and categories occur in less than 10% of the heats and are therefore bundled together into two aggregate variables, $SCR_{Aggr}$ and *Aggregate*, for the plant scrap representation and visual scrap representation, respectively.

Each of the three representations are further described in detail.

### 3.1.1. Steel Plant Scrap Yard System

The steel plant uses an internal coding system for the raw material types, which follow a logic that is based on where the raw material is sourced from, its quality, composition, and shape. The majority of the raw material types represent steel scrap. Although nickel granulates and anthracite, i.e., coal, have distinct coding, these raw materials will be referred to as scrap types well knowing that they are not scrap. The main reason is that the focus of this study is on the effect of steel scrap on the EE consumption of the EAF. The effect of alloying elements is not of interest. Furthermore, nickel granulates and anthracite represent a small fraction of the total amount of charged raw materials for all steel grades.

The composition of the scrap can vary from low-to high alloying elements such as carbon, chromium, nickel, and molybdenum. Furthermore, quality-hampering impurities such as copper and tin are prevalent in some scrap types that are considered of low quality. The scrap quality is also dependent on the confidence in the actual content of the scrap. In general, more reliability is put on quality from internal sourced scrap and scrap from other steel plants than on purchased scrap from municipal waste incineration plants. The shapes of scrap can vary from thin plates to bulky residuals from foundries and own arising scrap.

14 of the 33 scrap types were charged in less than 10% of the heats represented by the data. To reduce the number of variables in the numerical experiments, a variable was created as the sum of the charged weights of these scrap types. This variable is called $SCR_{Aggr}$, see also Figure 3.

The goal of including this representation in the numerical experiments is to investigate if the most granular representation of the available scrap, in the steel plant of study, is the most optimal with respect to the accuracy of the EE consumption prediction models.

### 3.1.2. Visual Categorization

The melting rate of scrap in the EAF is predominantly dependent on the surface area available to heat transferring media such as the hot heel and radiation from the arc plasma. By categorizing the scrap types according to their distinct physical shape, two benefits can be achieved. First, scrap types with similar shapes are expected to have closely related melting performance in the EAF due to their similar surface-area-to-volume ratios. Second, the number of variables will be reduced which is beneficial from a statistical modeling point of view.

The difference between the categories Internal1 and Internal2 is based on the apparent density and the difference between the categories Plate1 and Plate2 is based on the thickness of the plates. They were separated because a difference in apparent density and plate thickness will affect the melting time of the scrap pieces.

4 of 14 visual categories were charged in less than 10% of the heats represented by the data. As with the steel plant scrap representation, a variable was created as the sum of the charged weight of these scrap types to reduce the number of variables. This variable is called *Aggregate*.

### 3.1.3. Density Categorization

All scrap types have an estimated apparent density conducted by either established standards or by the steel plants engineers. The main goal of this scrap representation is to categorize each scrap type into four categories, which are based on the estimated apparent density, i.e., density interval. All scrap types have an estimated apparent density range or a lower bound apparent density value.

The following categories were created, based on ranges of reported apparent densities:

- **Light:** 0.3–1.0 ton/m$^3$
- **Light-Mid:** 0.56–1.0 ton/m$^3$
- **Mid:** 0.7–1.4 ton/m$^3$
- **Heavy:** >1.4 ton/m$^3$

Intuitively, scrap types with lower apparent densities should melt faster than scrap types with higher apparent densities. Since the density of the metal pieces can be assumed to be the same, the variation in apparent density is a measure of "porosity" of the scrap types. Higher "porosity" leads to more surface area available for heat transfer, which in its turn leads to higher melting rates and consequently shorted melting times.

## 3.2. Data Governing the EAF

### 3.2.1. Variable Selection

The selected variables are based on the discussion regarding the non-linearity of the EAF process, reported energy sources and energy sinks, the correlative relationships of the variables to the EE consumption, and the scrap variables governed by the three scrap representations (see Section 2 and Section 3.1). The selected variables from each type are shown in Table 2.

**Table 2.** The variables used in the models.

| Variables | Unit | Definition |
|---|---|---|
| Electrical Energy (EE) | kWh | The electrical energy consumption for the heat. |
| Total Weight | kg | The sum of all charged scrap types. |
| Tap-to-Tap time (TTT) | min | The time between the end of the tapping from the previous heat to the end of tapping of the current heat. |
| TCB2 | min | Time between start of heat until charging of the second basket. |
| Burner oil | kg | Total amount of oil added by burner. |
| Burner $O_2$ | $Nm^3$ | Total amount of oxygen added by burner. |
| $O_2$-lance | $Nm^3$ | Total amount of oxygen added by lance. |
| Injected carbon | kg | Total amount of carbon injection. |
| Lime and dolomite | kg | Total lime and dolomite added. |
| No. Charges | – | Total number of scrap baskets added. |
| $SCR_{101}$ | kg | Above 3 mm thick plate. Apparent density above 1.0 ton/$m^3$. |
| $SCR_{103}$ | kg | Thin plate, cuttings of rolled thin plate. Apparent density above 1.0 ton/$m^3$. |
| $SCR_{212}$ | kg | Heavy cuttings of internal scrap. Apparent density above 1.4 ton/$m^3$. |
| $SCR_{333}$ | kg | Anthracite (carbon). Apparent density 0.72 ton/$m^3$. |
| $SCR_{400}$ | kg | Purchased rebar and plates. Apparent density 0.56–1.0 ton/$m^3$. |
| $SCR_{407}$ | kg | Heavy melting mix (HM1). Apparent density above 0.7 ton/$m^3$. |
| $SCR_{409}$ | kg | Shredded scrap. Apparent density 1.0 ton/$m^3$. |
| $SCR_{412}$ | kg | Internal scrap. Apparent density above 1.4 ton/$m^3$. |
| $SCR_{416}$ | kg | Turnings. Apparent density above 0.4 ton/$m^3$. |
| $SCR_{451}$ | kg | Purchased scrap (HM1 and HM2). Apparent density above 0.7 ton/$m^3$. |
| $SCR_{452}$ | kg | Internal scrap. Apparent density above 1.4 ton/$m^3$. |
| $SCR_{455}$ | kg | Turnings. Apparent density above 0.4 ton/$m^3$. |
| $SCR_{492}$ | kg | Mixed internal scrap. Apparent density above 0.7 ton/$m^3$. |
| $SCR_{633}$ | kg | Si-rich plate. Apparent density 0.7–1.0 ton/$m^3$. |
| $SCR_{677}$ | kg | Incineration scrap. Apparent density above 0.5 ton/$m^3$. |
| $SCR_{686}$ | kg | Grinding swarfs and grinding swarf briquettes. Apparent density 0.4–1.0 ton/$m^3$. |
| $SCR_{698}$ | kg | Skulls. Apparent density above 0.7 ton/$m^3$. |
| $SCR_{699}$ | kg | Grinding swarfs and grinding swarf briquettes. Apparent density 0.3–1.0 ton/$m^3$. |
| $SCR_{711}$ | kg | Incineration scrap. Apparent density above 0.5 ton/$m^3$. |
| $SCR_{Aggr}$ | kg | Sum of scrap types charged in less than 10% of the heats. |
| Incineration | kg | |
| Heavy melting scrap (HM) | kg | |
| Plate1 | kg | |
| Plate2 | kg | |
| Internal1 | kg | |
| Internal2 | kg | Scrap representation based on the shape of the scrap type. See Section 3.1.2 and Figure 3. |
| Shredded | kg | |
| Swarfs | kg | |
| Turnings | kg | |
| Carbon | kg | |
| Skulls | kg | |
| Aggregate | kg | |
| Heavy | kg | |
| Mid | kg | Scrap representation based on the reported and estimated apparent density ranges or values of each respective |
| Light-Mid | kg | scrap type in the steel plant. See Section 3.1.3 and Figure 3. |
| Light | kg | |
| Hot heel | – | 1 if hot heel is present at the start of the heat, else 0. To account for the heat transfer by the hot heel. |
| Furnace shell number | – | An ordinary variable counting the number of heats since the last furnace barrel maintenance. |

### 3.2.2. Variable Batches

The variable batches are based on the variables motivated in Section 3.2.1 as a starting point. All 8 variable batches from this group can be seen in Table 3 and the variables in each variable group are shown in Table 4.

**Table 3.** The domain-specific variable batches. The variables present in each variable group are shown in Table 4.

| | Variable Batch | 1 | 2 | 3 | 4 | 5 | 6 | 7 | 8 |
|---|---|---|---|---|---|---|---|---|---|
| **Variable Group** | Base | x | x | x | x | x | x | x | x |
| | Plant scrap category | | | x | x | | | | |
| | Visual scrap category | | | | | x | x | | |
| | Density scrap category | | | | | | | x | x |
| | Furnace related | x | | x | | x | | x | |

The reason the scrap representations will not be used together in any of the variable batches is due to physical consistency. Besides being redundant, there is a physical logic tied to each scrap representation. Using a mixture of scrap representations will not indicate which scrap representation is the most optimal to use. One of the aims of this study is to investigate the best scrap representation with respect to the performance of the models on test data.

**Table 4.** Input variables for each variable group. There is a total of 48 input variables.

| Variable Group | Variables | No. Variables | Variable Group | Variables | No. Variables |
|---|---|---|---|---|---|
| **Base** | Total Weight<br>TTT<br>TCB2<br>Burner oil<br>Burner $O_2$ | 9 | | $SCR_{686}$<br>$SCR_{698}$<br>$SCR_{699}$<br>$SCR_{711}$<br>$SCR_{Aggr}$ | |
| | $O_2$-lance<br>Injected carbon<br>Lime and dolomite<br>No. Charges | | **Visual scrap representation** | Incineration<br>HM<br>Plate1<br>Plate2 | 12 |
| **Plant scrap representation** | $SCR_{101}$<br>$SCR_{103}$<br>$SCR_{112}$<br>$SCR_{333}$<br>$SCR_{400}$<br>$SCR_{407}$<br>$SCR_{409}$<br>$SCR_{412}$ | 20 | | Internal1<br>Internal2<br>Shredded<br>Swarfs<br>Turnings<br>Carbon<br>Skulls<br>Aggregate | |
| | $SCR_{416}$<br>$SCR_{451}$<br>$SCR_{452}$<br>$SCR_{455}$ | | **Density scrap representation** | Heavy<br>Mid<br>Light-Mid<br>Light | 4 |
| | $SCR_{492}$<br>$SCR_{633}$<br>$SCR_{677}$ | | **Furnace related** | Hot heel<br>Furnace shell number | 2 |

### 3.3. Data Treatment

### 3.3.1. Purpose

To ensure the reliability and validity of a statistical model, the data which is used to create the model must be treated. The reason is because statistical models adapt their coefficients solely based on data, as opposed to physical models, which have pre-determined coefficients. By including data

that is of low quality, the statistical model will inherit that quality when making predictions on new, previously unseen, data. In general, data treatment is a double-edged sword. On the one hand, a model should be able to predict well on any future data. On the other hand, all data sets contain data points that represent extreme cases. Any statistical regression model will predict these extreme cases with a low accuracy, since the coefficient adaptation algorithm is based on minimizing the error on the entire data set included in the training phase. Any extreme case receives a lower priority due to its rarity. Hence, a successful modeling effort strikes a balance between these two opposing effects.

Data treatment methods can be divided into two categories, domain-specific methods and statistical methods. The two disparate data treatment methods will be described further.

### 3.3.2. Domain-Specific Methods

This method uses the knowledge and experience in the domain from which the data originates. Domain-specialization and manual treatment of the data are required. This can be both expensive and time consuming but the end-result, i.e., the cleaned data, is expected to be of higher quality than using pure statistical outlier detection algorithms, which do not adhere to the domain-specific considerations. In the scope of the EAF process, an example of a domain-specific treatment is the removal of instances of data that are improbable. If the registered charged content of scrap is 190 t while the maximum capacity of the furnace is 120 t, then that instance should be removed. Another example is the removal of heats that are not part of regular production. For example, testing the effects of new scrap types, or delivery batches.

### 3.3.3. Statistical Methods

Statistical data treatment methods refer to algorithms that identify anomalies in the distributions in data sets. The algorithms are purely mathematical and do not take into account any domain-specific reasons behind anomalies or domain-specific relations between the variables. The advantage of this method is that it is easy to apply and does require very little, if any, domain-specific knowledge. However, applying outlier detection algorithms to multiple variables will potentially remove most of the data. A simple example is the various types of scrap types used in production, which are prone to extreme (very high or very low) values since some scrap types are not available at all times. Applying a statistical cleaning algorithm to all scrap types will omit most of the data. Hence, statistical treatment methods must be used with caution and only on well-selected variables that are expected to impact the predictions dramatically. For example, total charged scrap weight and TTT.

Tukey's fences [19], which is based on the interquartile range of a distribution, will be used in the numerical experiments. The method removes data points that are present outside the range defined as:

$$q_1 - \epsilon(q_1 - q_3) \leq x_j \leq q_3 + \epsilon(q_1 - q_3) \tag{5}$$

where $q_1$ and $q_3$ are the first and third quartiles of variable $j$, respectively. $\epsilon$ is a pre-specified constant indicating how far out the outlier must be before being cleaned. $\epsilon = 3.0$ will be used in the numerical experiments which removes extreme outliers [19].

Since this method is based on the quartiles of the distribution under consideration, it is less sensitive to skewed distributions compared to cleaning by omitting data points outside of $\pm 3\sigma$ from the mean. This inherent characteristic of Tukey's fences is advantageous, since most variables governing the EAF process are non-Gaussian. See Figure 4 for examples.

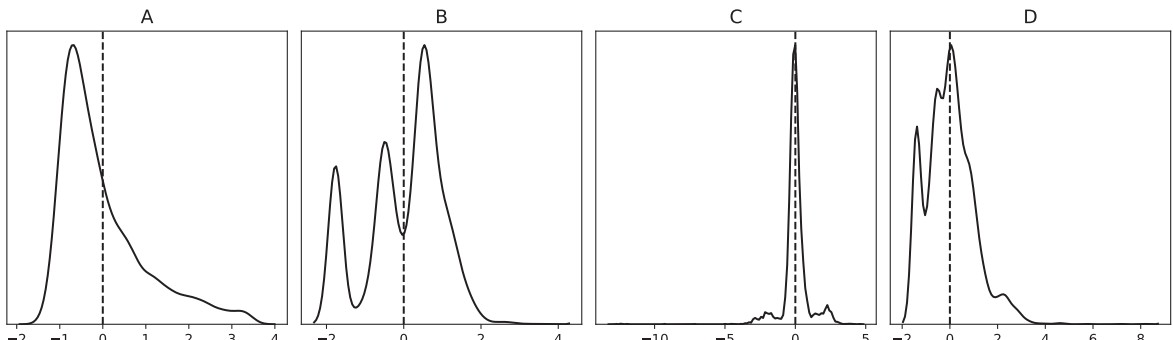

**Figure 4.** The distributions for four variables governing the EAF under study highlighting the absence of the Gaussian distribution. All values are normalized, and the dashed lines indicate the mean values. (**A**): TTT. (**B**): Charged weight of internal scrap. (**C**): Total charged weight of raw materials. (**D**): Charged weight of shredded scrap.

### 3.3.4. Applied Data Treatments

Due to the opposing effects of data treatment as mentioned earlier, it is impossible to know which approach strikes a good balance between model generalizability and model accuracy. Therefore, four different data treatment approaches will be used in the modeling to investigate the influence of the different approaches.

The first data treatment approach was conducted by a senior engineer at the steel plant. This data treatment was done by manually inspecting each row of the data set and flagging rows which contained values that were not consistent with the data instance as a whole. This data treatment is a combination of domain-expertise and some subjectivity of the senior engineer. However, because the data is inherently coupled with the steel plant it originates from, using on-plant experts is critical to a successful data treatment operation. This data treatment is referred to as *Expert*.

For the following three data treatment approaches, two filter steps were applied to remove unrealistic heats with respect to events in time. The first filter removed heats where the timestamp of charging the second heat was negative respect to the start of the heat. The second filter removed heats where the charging of the second basket occurred after the heat ended. Applying these filters were enough to remove all unrealistic heats.

The second data cleaning approach was conducted by the authors of this study which was based only on domain-specific knowledge. Two cleaning steps were applied. The first cleaning step removed heats with TTT at, or above, 180 min since these heats are likely experiencing a longer delay in the process or a scheduled stop. Usually, the TTT is aimed at 60–70 min. The second cleaning step removed heats with a Total charged weight at, or above, 141 ton. This is a limit set by the steel plant for abnormally large charge weights. This data treatment approach is referred to as *Domain-specific*.

The third data treatment used Tukey's fences to remove clear outliers, see Section 3.3.3. Tukey's fences were calculated and applied to each of the following input variables using the training data: Total Weight, TTT, Time until Charging of Basket 2 (TCB2), Burner $O_2$, Burner oil, $O_2$-lance, and Injection carbon. Each 'fence' was then applied to the training and test data, respectively. This data treatment approach is referred to as *Tukey*.

The fourth data treatment approach used the second and third data treatment approaches, in order. This data treatment approach is referred to as *Domain-specific Tukey*.

Each of the four described data treatment approaches will be used in the modeling to investigate which one of the data treatments is the most optimal for a model applied in practice.

*3.4. Modeling the EE Consumption*

3.4.1. Statistical Modeling Frameworks

To consider the non-linearity of the EAF process, a non-linear statistical modeling framework must be used. The numerical experiments in this study will use two different non-linear statistical modeling frameworks; ANN and RF. ANN has been used in a previous article using the same methodology as described in this paper [2]. RF will be used to broaden the scope of models used to predict the EE consumption using the same methodology. Furthermore, SHAP interactions, an interpretable machine learning algorithm that calculates the interactions between the input variables with respect to the output variable can be used for RF models [20]. This is not the case for ANN models where only regular SHAP values can be used. SHAP will be further explained in Section 3.5.1.

**Artificial Neural Networks:** This model framework uses a fully connected network of nodes to make predictions [21]. The first layer, which is known as the input layer, receives the values from the input variables. The values are then propagated through the intermediate layers, which are known as hidden layers, to the last layer. The last layer is the output layer where the prediction is made. See Figure 5 for an illustration of an arbitrary ANN model.

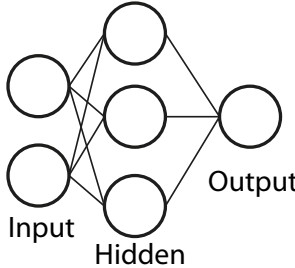

**Figure 5.** An Artificial Neural Network (ANN) for predicting an output value based on two input values [2]. It has one hidden layer with three nodes. The lines between the nodes illustrate that the ANN is fully connected and the forward flow of calculations in the network.

By changing the number of nodes in the hidden layers and the number of hidden layers, one can alter the complexity of the model. Increasing the number of hidden layers and nodes increase the complexity and enable the model to learn more complicated relations between the input variables and the output variable.

In the output layer, and in each of the hidden layers, each node multiplies a weight value with each of the value propagated by the previous layer. The resulting weight-value factors are summed together. Mathematically, this process can be expressed as:

$$s_j = \sum_{i=1}^{P} w_i \cdot x_i \tag{6}$$

where $j$ is the $j$:th node in the current layer and $P$ is the number of nodes in the preceding layer.

Each value, $s_j$, is then fed into an activation function, and the resulting value is propagated to the next layer in the network. Two commonly used activation functions are the hyperbolic tangent (tanh) and the logistic sigmoid.

During the training phase, the weights are updated in the direction of minimizing the overall loss of the predictions on the training data. Since the output variable is connected to the input variables by the network weights, it is possible to mathematically express the loss as a function of the network weights. Finding an optimal local minimum, with respect to the overall loss, in the weight space requires a sophisticated algorithm. These algorithms are known as gradient-descent algorithms as their function is to descent to the most optimal local minima in loss space [21].

Given enough hidden layers and nodes, an ANN can learn any complex relationship between variables even though the relationships are not valuable for prediction purposes. This overfitting

phenomenon can be reduced by splitting the training data into two sets. The first set of data is used to adapt the weights while the other set is used to calculate the loss after each weight update.

**Random Forest:** This statistical modeling framework is a model made of two, or more, decision trees. See Figure 6 for an illustration of a simple decision tree for prediction purposes. The RF model framework was first reported by L. Breiman [22]. RF belongs to the statistical model group known as ensemble models, which is a group of statistical models that is made up of two or more models that when combined, aim to increase the prediction accuracy.

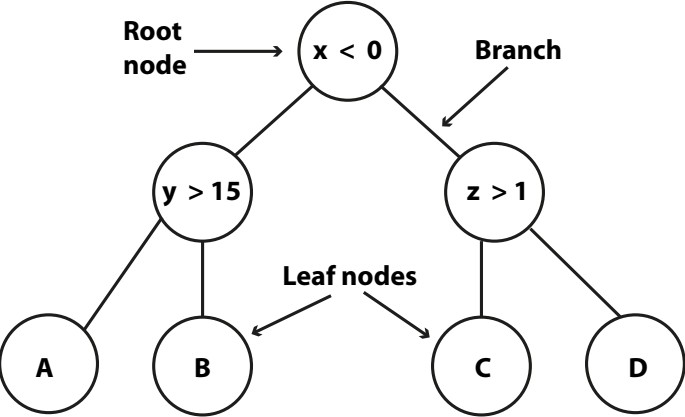

**Figure 6.** A simple decision tree sorting points on the $\{x, y, z\}$ coordinate system. Points satisfying the conditions on each node proceed on the left branch, the rest proceed to the right branch. The points $A = \{0, 42, 5\}$, $B = \{-10, -10, -10\}$, $C = \{31, 4, 0\}$, and $D = \{2, 4, 61\}$, are sorted in the decision tree.

In an RF model, each decision tree is trained on a sub-sample of the complete training data set. This sub-sample is drawn *with replacement* from the complete training data set, a process known as bootstrapping. By training each decision tree on a sub-sample of the training data set the overfitting of the RF model is reduced, since each decision tree becomes specialized on one segment of sample space. Furthermore, the optimal split for the next branch from each node in the decision tree is selected using a random selection of a pre-specified number of the total available input variables. This procedure also reduces overfitting since each decision tree now has a higher probability of being diverse with respect to the other trees in the model. Using many trees created by a random selection of features and data points, RF type models have proven to converge such that overfitting does not become a problem [22]. In the prediction phase of an RF model, each decision tree predicts the value of the output variable. In a prediction of regression type, i.e., when predicting continuous values, the prediction by the RF model is the average of the predictions from all decision trees. The prediction of one data instance, $x_k$, in an arbitrary RF model is illustrated in Figure 7.

To optimize an RF model for the task at hand, i.e., improve the accuracy, one must search for an optimal combination of hyper-parameters. The most important hyper-parameters are the maximum tree depth, i.e., number of splits from the root node, the number of decision trees, and the maximum number of features used to find the optimal condition when splitting a node [23].

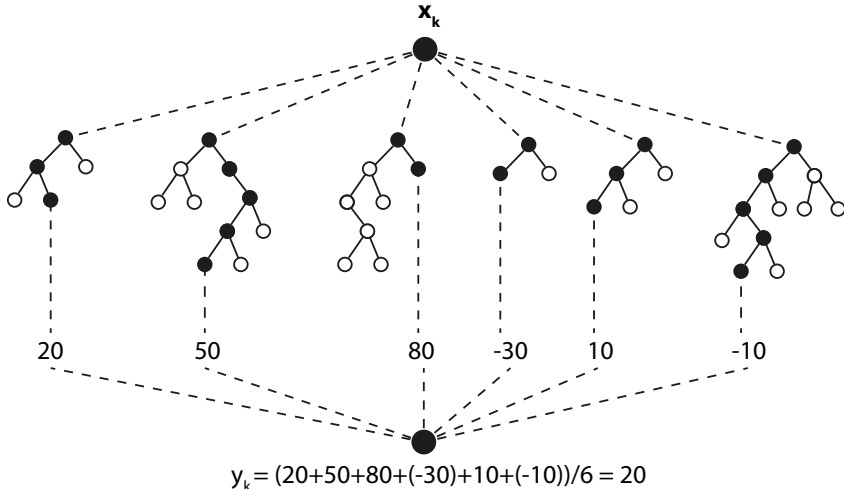

**Figure 7.** An arbitrary RF model consisting of 6 decision trees. The prediction, $y_k$, of the data instance, $x_k$, is determined by averaging the sum of the outputs from the decision trees. The filled nodes show the path $x_k$ has taken in each decision tree.

### 3.4.2. Parameter Optimization

The variations of parameters for each statistical model framework, RF and ANN, can be seen in Tables 5 and 6, respectively. Each combination of parameters represents one model type. The data cleaning strategies and variable batches were included as additional, model framework independent, parameters. By including the cleaning strategies as a parameter, it is possible to find out which cleaning strategy achieves the most optimal trade-off between accuracy and data omission. By including the variable batches as a parameter, it is possible to find the model with a relative high accuracy but with the least amount of input variables. One should always abide to the concept of model parsimony, which is to select the simplest model among a set of models with next to the same performance.

**Table 5.** Parameter combinations used for the RF models. Each value is separated by a comma. $m$ is the number of input variables. Each combination of parameters represents one model type.

| Parameter | Variations | #Combinations |
| --- | --- | --- |
| Number of trees | 10, 30, 50, 70, 90, 110, 130, 150, 170, 190, 210, 230, 250 | 13 |
| Max tree depth | 2, 3, 4, 5, 6, 7, 8, 9, 10, Unlimited | 10 |
| Max features in split | $v, \sqrt{v}$ | 2 |
| Cleaning strategies | See Section 3.3 | 4 |
| Variable batches (domain-specific) | See Section 3.2.1 | 8 |
| | **Total:** | 8320 |

**Table 6.** Parameter combinations used for the ANN models. Each value is separated by a comma. The topology (z) and (z,z), indicate one and two layers with z nodes in each layer, respectively. Each combination of parameters represents one model type.

| Parameter | Variations | #Combinations |
|---|---|---|
| Activation function | Hyperbolic tangent function, Logistic sigmoid | 2 |
| Learning rate | 0.1, 0.01, 0.001 | 3 |
| Topology | (z) and (z,z) in z ∈ 1, 3, ..., 29 | 30 |
| Cleaning strategies | See Section 3.3 | 4 |
| Variable batches (domain-specific) | See Section 3.2.1 | 8 |
| **Total:** | | 5760 |

### 3.4.3. Selection of Training and Test Data

Selecting the test data from a random sub-sample of the complete data makes the training and test data become chronologically intertwined. This is a shortcoming because it does not reflect the practical purpose of a statistical model predicting the EE of an EAF [1]. From a process perspective, a statistical model will predict on heats that are from a future point in time with respect to the heats whose data have been used to adapt the parameters of the model, i.e., training data. To account for this shortcoming, the test data will be selected in chronological order from the training data. The test data will be all heats produced from 1st of February 2020 to the 28th of February 2020 and the training data will the heats produced from the 10th of November 2018 through January 2020. The start date of the training data was selected based on a furnace upgrade that was completed on the 9th of November 2018. This amounted to 4032 training data points and 263 test data points before data treatment. The heats in the training data and test data will be referred to as *training heats* and *test heats*, respectively.

### 3.4.4. Model Performance Metrics

The performance of the models will be compared using two fundamental metrics. These are the coefficient of determination, $R^2$, and the regular error metric.

The adjusted-$R^2$ value should be used instead of the regular $R^2$ value when comparing models that use different number of input variables. The reason is because each additional input variable increases the $R^2$-value when the number of data points is fixed [24]. The adjusted-$R^2$ value can be calculated as follows:

$$\bar{R}^2 = 1 - (1 - R^2)\frac{n-1}{n-v-1} \tag{7}$$

where $n$ is the number of data points, is the number of input variables, and $R^2$ is the regular R-square value.

The regular error metric was chosen in favor of the absolute error metric, because in a practical context, an overestimated prediction of EE is vastly different from an underestimated prediction of EE. The regular error metric can be defined as follows:

$$E_i = y_i - \hat{y}_i \tag{8}$$

where $y_i$ is the true value, $\hat{y}_i$ as the predicted value, $i \in 1, 2, \ldots, n$, and $n$ is the number of data points. Using all the data points under consideration, the standard deviation, mean, minimum, and maximum, error values are defined the ordinary way.

The EE consumption, and therefore the unit of error, will be expressed in kWh/heat rather than in kWh/t tapped steel. The main reason for this choice is that the former varies only by consumed EE while the latter varies both by consumed EE and the yield. This is a more challenging problem

as it requires the statistical model to adapt to both the consumed EE as well as the yield, which is defined as the weight of tapped steel divided by weight of charged scrap. Furthermore, the tap weight is dependent on factors such as the slag created by oxidation in the process and on the total amount of dust generated. These two examples are, in turn, affected by process times, amount of oxygen injected, additives, and the charge mix.

### 3.4.5. Model Selection

Each model type, which is defined by one of the parameter combinations shown in Tables 5 and 6, will be instantiated 10 times. The reason for this is to investigate the stability of each model type and to reduce the impact of randomness, which is prevalent in each of the statistical model frameworks. In RF, the randomness is introduced by random selection of data points for each tree and the random selection of input variables to determine each split. In ANN, the randomness is partly governed by the random selection of the initial values of the network weights.

The aggregate statistical metrics based on the 10 model instances of each model type are presented in Table 7.

**Table 7.** The adjusted-$R^2$ and error metric variants that are used to evaluate the performance of the aggregated model instances.

| Symbol | Definition |
|---|---|
| $\bar{R}_{\mu}^2$ | Mean adjusted R-square of the 10 model instances on the test data |
| $\bar{R}_{\sigma}^2$ | Standard deviation of adjusted R-square of the 10 model instances on the test data |
| $\bar{R}_{min}^2$ | Minimum adjusted R-square of the 10 model instances on the test data |
| $\bar{R}_{max}^2$ | Maximum adjusted R-square of the 10 model instances on the test data |
| $\Delta_{\mu}$ | Mean error of the mean error of the 10 model instances on the test data |
| $\Delta_{\sigma}$ | Standard deviation of the mean error of the 10 model instances on the test data. |
| $\Delta_{min}$ | Minimum error of the mean error of the 10 model instances on the test data. |
| $\Delta_{max}$ | Maximum error of the mean error of the 10 model instances on the test data. |

To determine the stability of the models, which is influenced by the underlying randomness, the idea behind the model selection criteria was to keep the difference between $\bar{R}_{max}^2$ and $\bar{R}_{min}^2$ as low as possible. Hence, the algorithm for selection the best model type of each variable batch can be expressed as follows:

1. Select all models that pass the following condition: $\bar{R}_{max}^2 - \bar{R}_{min}^2 \leq 0.05$.
2. Select the model(s) with the highest $\bar{R}_{\mu}^2$. If the number of models exceeds 1, then proceed to the next step, otherwise select the one model.
3. Select the model with $min(\bar{R}_{max}^2 - \bar{R}_{min}^2)$.

The adjusted-$R^2$ was chosen to determine the stability of the models because it indicates the goodness of fit. The other error metrics, based on the model error, do not relate to goodness of fit.

Using the above algorithm, the number of models will be reduced to 8, which is equal to the number of variable batches. From this model subset, the model with the fewest number of input variables will be selected, given that more than one model have the same $\bar{R}_{\mu}^2$-value. This ensures that the model selection adheres to the concept of model parsimony, i.e., that the simplest model, which is the model with least number of input variables, is selected when more than one model have the same performance.

### 3.5. Model Evaluation and Analysis

### 3.5.1. Shapley Additive Explanations (SHAP)

Any prediction by the statistical model, $f$, can be explained by a linear combination of the contributions by all input variables:

$$f(x) = \phi_0 + \sum_{i=1}^{M} \phi_i \tag{9}$$

where $\phi_0$ is the contribution when information about the variables are not present. In practice, this will always be the mean value of all prediction of the statistical model. This is sensible since the mean value is always the most optimal choice if complementary information is lacking.

Each SHAP value, $\phi_i$, can be calculated using the following expression [9]:

$$\phi_i(f, x) = \sum_{Z \subseteq \hat{Z} \setminus \{i\}} \frac{|Z|!(M - |Z| - 1)!}{M!} [f_x(Z \cup \{i\}) - f_x(Z)] \tag{10}$$

where $Z$ is a subset of the set of all input variables, $\hat{Z}$, and $M$ is the total number of input variables. $\hat{Z} \setminus \{i\}$ is the set of all input variables excluding variable $i$ and $Z \cup \{i\}$ is a subset of the set of all input variables including variable $i$. Hence, each SHAP value, $\phi_i$, represents the average contribution by an input variable on the output variable of all combinations in which that input variable is presented to the statistical model, $f$.

The number of calculations required to calculate Equation (10) scales drastically when the number of input variables and data points increases. To reduce the number of calculations, one can use an approximative method known as the Kernel SHAP method, which assumes variable independence and model linearity. Assuming variable independence means that Kernel SHAP will produce value combinations that are unrealistic for variables that are dependent. For example, high amount of burner oil will be matched with low oxygen through the burners. The linearity assumption is analogous to linear approximation of functions in mathematics where the space in the vicinity of a data point is assumed to be linear. Furthermore, Kernel SHAP is model independent which means that it can be applied to any supervised statistical model. It has previously been used to analyze an ANN predicting the EE of an EAF [10].

RF enables the use of the Tree SHAP method, which is adapted to tree-based statistical models. Tree SHAP can calculate the exact SHAP values because it does not assume variable independence. Furthermore, the algorithm is computationally efficient compared to the regular SHAP method [20], which means that the SHAP values can be calculated within a reasonable timeframe. By not assuming variable independence, Tree SHAP adheres to the true behavior of the model as opposed to Kernel SHAP, which uses a simplified representation of the original model. This is important since the aim of SHAP is to explain the behavior of the statistical model as accurately as possible.

Tree SHAP also provides the ability to use SHAP interaction values, which calculates the interaction effects between the input variables as contributions to the prediction [25]. Instead of receiving one SHAP value per input variable and prediction, the interaction value calculation provides an $MxM$ matrix per prediction, where $M$ is the number of input variables. The diagonal of the matrix contains the main interaction values, $\phi_{i,i}$, which are the contributions by each input variable without the influence of the other input variables. The upper and lower triangles of the matrix contain the one-to-one interaction values, $\phi_{i,j}$, which show the contributions to the prediction by each input variable pair. The SHAP interaction value is equally split such that $\phi_{i,j} = \phi_{j,i}$, which means that the total contribution of the input variable pair $(i, j)$ is $\phi_{i,j} + \phi_{j,i}$.

The SHAP interaction values can be calculated by [25]:

$$\phi_{i,j}(f, x) = \sum_{Z \subseteq \hat{Z} \setminus \{i,j\}} \frac{|Z|!(M - |Z| - 2)!}{2(M - 1)!} \nabla_{i,j}(Z) \tag{11}$$

where

$$\nabla_{i,j}(Z) = f_x(Z \cup \{i, j\}) - f_x(Z \cup \{i\}) - f_x(Z \cup \{j\}) + f_x(Z) \tag{12}$$

The SHAP interaction values can be expressed as the regular SHAP values as follows:

$$\phi_i = \phi_{i,i} + \sum_{i \neq j}^{M} \phi_{i,j} \tag{13}$$

Henceforth, the SHAP interaction values relate to the model prediction as:

$$\sum_{i=0}^{M} \sum_{j=0}^{M} \phi_{i,j}(f, x) = f(x) \tag{14}$$

This means that the SHAP interaction values are a more granular representation of the contribution by each input variable on the prediction as opposed to regular SHAP values. On the one hand, the main interaction values provide us with the interaction of each input variable unaffected by the influences of the other input variables. On the other hand, the one-to-one interaction values between the input variables provide us with information on how the interaction between the input variables adds to the prediction contribution.

The aim of showing the main interaction effects on EE by the input variables is to investigate if the models adhere to the underlying relationships between the input variables and the output variables. One must bear in mind that this analysis is univariate with respect to the output variable. Hence, it is not possible to draw any conclusions regarding the intra-relationships between the input variables that together affect the output variable.

There is also one concept known as SHAP feature importance. It is defined as the mean absolute value of the regular SHAP values. See Equation (15).

$$FI_j = \frac{1}{n} \sum_{i=1}^{n} \phi_j^i \tag{15}$$

where $FI_j$ is the SHAP feature importance for variable $j$ and $n$ is the number of data points.

SHAP feature importance measures the global importance of each input variable and is a more trustworthy measure of feature importance than the traditional permutation-based feature importance. The main reason is that SHAP feature importance is rooted in solid mathematical theory while permutation-based feature importance is based on the empirical evidence provided by random permutations.

In the numerical experiments, the package *shap*, with the method Tree Explainer, in Python will be used to calculate SHAP. The method 'tree path dependent' will be used since it adheres to the variable dependence among the input variables. The software and hardware used in the numerical experiments can be seen in Appendix A, Tables A1 and A2.

3.5.2. Correlation Metrics

Two different correlation metrics will be used to investigate the intra-correlation between input variables as well as the correlation between the input variables and EE, i.e., the output variable. By studying the resulting correlation values with domain-specific knowledge, it is possible to verify the connection between the model prediction and the input variables. After all, the intra-correlation between input variables and their respective correlations with the output variable is what the model learns during the training phase. The two correlation metrics are explained further.

**Pearson correlation:** The Pearson correlation metric that can only detect linear relationships between two random variables [26]. It assumes values between $-1$ and $1$, where the former is a perfect negative relationship between the variables and the latter is a perfect positive relationship, i.e., the variables are identical. A value of 0 indicates that the variables have no relation.

The Pearson correlation coefficient is defined as follows:

$$\rho_{X,Y} = \frac{covariance(X,Y)}{\sigma_X \sigma_Y} \tag{16}$$

where $\sigma_X$ and $\sigma_Y$ are the standard deviations of the two random variables $X$ and $Y$, respectively.

**Distance correlation (dCor):** Although dCor cannot distinguish between positive and negative correlative relationships between variables, it is able to detect non-linear relationships between variables [27]. This is important since some variables governing the EAF process have a non-linear relationship to EE. By using dCor and Pearson in tandem, it is possible to get a clearer picture of the relationships between the variables governing the statistical models.

The mathematical expression for dCor has the same form as the Pearson correlation coefficient:

$$dCor(V_1, V_2) = \frac{dCov(V_1, V_2)}{\sqrt{dVar(V_1)dVar(V_2)}} \tag{17}$$

where $dVar(V_1)$ and $dVar(V_2)$ are the distance variance of the random variables $V_1$ and $V_2$, respectively. $dCov(V_1, V_2)$ is the corresponding distance covariance. The square root of $dVar(V_1)$ and $dVar(V_2)$ gives the distance standard deviations.

dCor assumes values between 0 and 1, where the former indicates that the variables are independent, and the latter indicates that the variables are identical. dCor has been used previously when evaluating the variables governing statistical models predicting the EE of the EAF [2,10].

### 3.5.3. Charge Types

Three distinctly different charge types, i.e., scrap recipes, with different tramp element contents will be used in the analysis to exemplify the contribution to EE by the scrap type or scrap category for each of the chosen charge types. The first charge type, denoted A, has the lowest level of tramp elements of the charge types used in the steel plant. It therefore requires higher amounts of scrap types where the amount of tramp elements such as Cu and Sn are well-known. Examples are residual scrap from the forging mill and purchased scrap from other steel mills. The second charge type, denoted B, does not have as strict requirements as charge type A. Hence, a higher amount of scrap types with lower qualities can be used. A typical lower quality scrap type is heavy melting scrap (HM) which can have relatively high amounts of tramp elements. The last charge type, denoted C, can have higher contents of tramp elements. Hence, purchased scrap from other steel plants and residual scrap are used to a lesser extent.

It is important to note that alloying elements promoting the desired steel properties such as Ni, Mo, and Cr come either from own arising scrap or from purchased scrap with low level of impurities. In the cases the internal scrap is not used, pure alloying elements such as Ni-granulated must be used instead. This is a more expensive route with respect to the total cost of raw materials.

## 4. Results and Discussion

### 4.1. EE consumption Models

$\bar{R}_\mu^2$-values for all variable batches and data cleaning strategies are shown in Figure 8. Consistency throughout the variable batches can be observed for all four cleaning strategies when the RF model framework is used. The consistency among the variable batches means, for the RF model framework that the deciding factor for the performance is the variable batch and not the hyper-parameters of the statistical model framework. This is a wanted outcome since the performance of a model should mainly be dependent on factors stemming from the application domain and not based on an optimization of parameters in an abstract framework.

Variable batches (VB) 1 and 2 produce models with the lowest $\bar{R}_\mu^2$-values. These are also the VB that *do not* use scrap representation variables. This provides evidence that scrap types are relevant factors when determining the EE consumption of the EAF and that scrap should not be treated collectively using only the total weight of all charged scrap.

The best performing models are those from VB 5 and 6 which use categories from the visual scrap representation. This provides evidence that a categorization based on scrap shapes is an optimal approach when creating a statistical model predicting the EE in the steel plant under study.

The performance of the models using the visual scrap representation are, performance-wise, followed by the models using the plant scrap representation (VB 3 and 4) and the density scrap representation (VB 7 and 8), respectively. Essentially, this indicates that a too fine or a too coarse representation of the charged scrap are sub-optimal for a statistical model predicting the EE. The steel plant scrap representation has numerous scrap types that contain the same scrap with respect to shape and dimension. The only difference is varying alloying content from Ni, Cr, and Mo, which does not significantly affect the melting time.

The coarse representation is based on the apparent density of the scrap, which does not take into account the shape of the scrap. Scrap shapes are closely related to the area-to-volume ratio, which is the strongest factor determining the melting time of scrap in the EAF since the stirring in the EAF is low during a large part of the melting phase. For example, HM has the same apparent density as skulls, which consist of bulky mixtures of solid slag and metal that takes long time to melt. Likewise, thin and thick plate have similar densities but different area-to-volume ratios.

The consistency among the four sets of VB between the four cleaning strategies for both statistical model frameworks further strengthens the evidence regarding the effects of the chosen scrap representations on the predictive performance of the models.

The performance and meta data of the best models and meta data from each cleaning strategy and model framework are shown in Table 8. In general, the ANN models perform similarly or better than the RF models with regards to the $\bar{R}_\mu^2$-values. However, the ANN models always have a smaller mean error and standard deviation of error, i.e., $\Delta_\mu$ and $\Delta_\sigma$.

With regards to the modeling meta data, the ANN and RF models had the same VB for the best models on the data from each cleaning strategy. This is also a wanted outcome based on the same reasoning as before regarding the importance priority between the domain-specific factors and the abstract model-based factors.

The total amount of cleaned data points was 25.8 percentages higher for the *Expert* cleaning strategy compared to the *Domain-specific* cleaning strategy. The $\bar{R}_\mu^2$-values only increased slightly using the *Expert* cleaning strategy; 0.035 and 0.039 for the RF and ANN models, respectively. Using the statistical cleaning method Tukey's fences, the amount of data cleaned were 11 and 13.3 percentages higher than the *Domain-specific* cleaning strategy. As opposed to the *Expert* cleaning strategy, the $\bar{R}_\mu^2$-values were worse for the models involving Tukey's fences. *Tukey* reported reduced $\bar{R}_\mu^2$-values of 0.049 and 0.077 for the RF and ANN models, respectively. For the *Tukey-Domain-specific* the reduction in $\bar{R}_\mu^2$ were 0.053 and 0.078, respectively. These results lead to two important findings. First, the usage of statistical cleaning heuristics results in a model performance that is sub-par to models using data cleaned by the usage of domain-specific knowledge; the *Expert* and *Domain-specific* cleaning strategies. Second, using data cleaned by an *Expert* yields models with the best performance, which illuminates the importance of knowledge about the specific EAF operations one intends to model. However, the large relative percentage of data loss using the *Expert* cleaning strategy (34.2%) as opposed to the *Domain-specific* cleaning strategy (10%) tilts the chosen cleaning strategy in favor of the latter since the data loss percentage directly relates to the percentage of future heats the model can predict on. This finding is closely tied to the practical usefulness of the model.

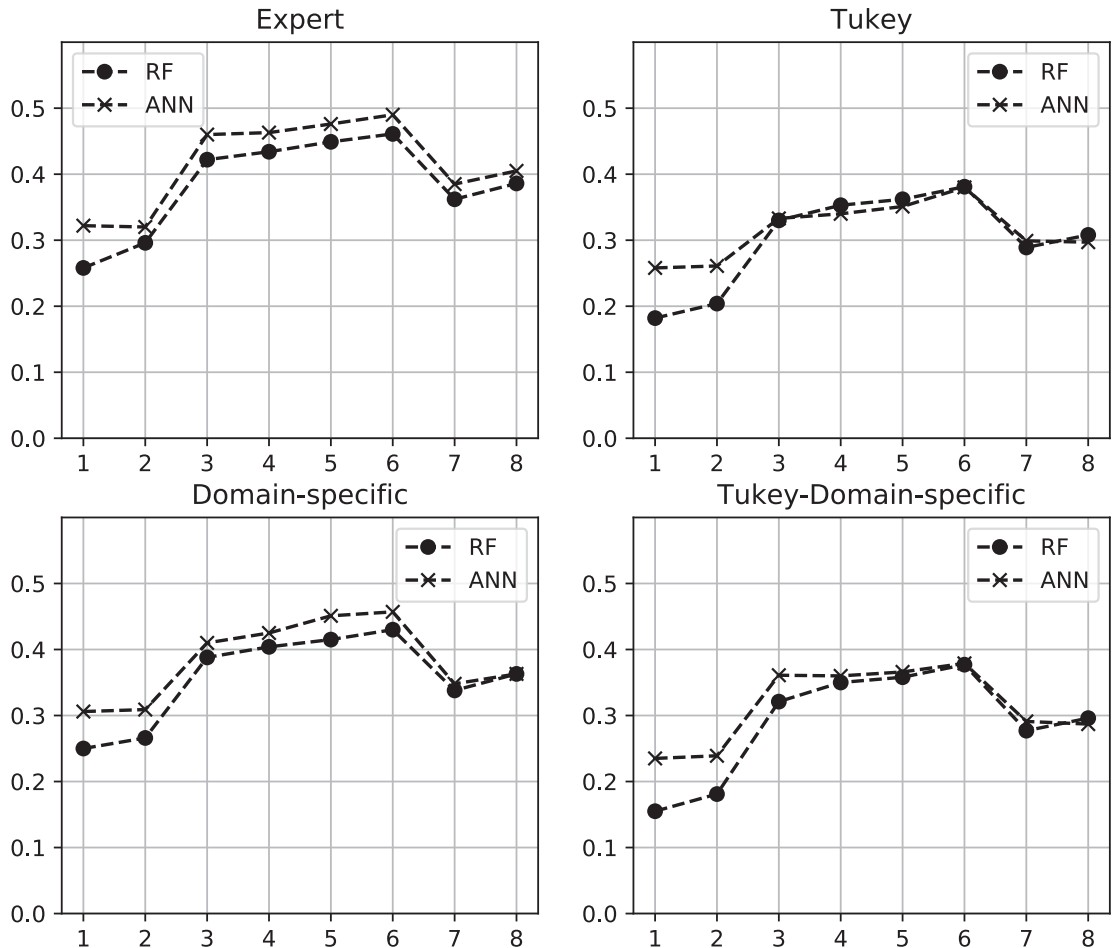

**Figure 8.** $\bar{R}_{\mu}^2$-values for each variable batch (VB 1-8 on the abscissa) and cleaning strategy. **VB 1–2:** Without scrap representation. **VB 3–4:** Steel plant scrap representation. **VB 5-6:** Visual scrap representation. **VB 7–8:** Density scrap representation.

**Table 8.** The performance and meta data of the best models, and the meta data for the four cleaning strategies. The values in parentheses show the values from the ANN models.

| Cleaning Type | Expert | Domain-Specific | Tukey | Tukey-Domain-Specific |
|---|---|---|---|---|
| Train/test split | 2571/187 | 3625/241 | 3179/212 | 3089/207 |
| % cleaned train/test | 36.2%/28.9% | 10.1%/8.4% | 21.2%/19.4% | 23.4%/21.3% |
| % total cleaned data | 35.8% | 10.0% | 21.0% | 23.3% |
| Variable batches | 6 (6) | 6 (6) | 6 (6) | 6 (6) |
| No. Variables | 21 (21) | 21 (21) | 21 (21) | 21 (21) |
| $\bar{R}_{\mu}^2$ | 0.461 (0.490) | 0.430 (0.457) | 0.381 (0.380) | 0.377 (0.379) |
| $\Delta_{\mu}$ (kWh/heat) | 408 (238) | 369 (209) | 252 (214) | 239 (126) |
| $\Delta_{\sigma}$ (kWh/heat) | 1938 (1863) | 2173 (2063) | 2062 (1995) | 2042 (1976) |
| $\Delta_{min}$ (kWh/heat) | −4496 (−5316) | −7064 −(7756) | −5061 (−5464) | −5079 (−4262) |
| $\Delta_{max}$ (kWh/heat) | 7113 (5699) | 7833 (7735) | 7810 (7608) | 7942 (7162) |

### 4.2. Analysis of the Selected Model

The selected model for the analysis is the RF model with a variable batch 6 using domain-specific cleaning. This model was selected based on the relatively high $\bar{R}_\mu^2$-value (0.457) while still keeping 90% of the data compared to the model using the expert cleaning type, which provided the highest $\bar{R}_\mu^2$-value (0.490) but only keeping 64.2% of the data. Hence, the chosen model can be used in 90% of future heats given that the test data and training data come from the same distribution.

The main interactions of the variables Plate1, Internal1, HM, and Shredded on the EE consumption and the charged weight distributions of these scrap categories for each of the charge types A, B, and C, can be seen in Figure 9.

The EE contribution by Plate1 on charge type A is lower than for B and C. However, charge type B can have both a positive and negative EE contribution since the densest part of the distribution is present in the steepest EE interaction change for Plate1. Hence, it is hard to conclude whether charge type B gets a similar contribution as does charge type A or charge type C. Charge type C is commonly using zero, or next to zero, amount of Plate1. Hence there is a large contribution to EE by Plate1 for this charge type. The EE contribution by Internal1 is slightly lower for charge type A than charge types B and C. The EE contribution by HM on steel type A is higher than for charge types B and C, the latter two of which are charged similarly across all heats. For the Shredded scrap category, all charge types receive similar contributions to EE.

Based on these highlights, it is expected that charge type A has the lowest EE consumption, that charge type C has the highest EE consumption, and that the EE consumption of charge type B is in-between those of type A and type C. The following EE consumption, relative to the EE consumption by charge type A, was obtained based on the data from the heats used to create Figure 9: charge type B = 0.99 and charge type C = 1.02.

On the other hand, charge type B requires slightly less EE consumption on average than charge type A (1.00). This is likely because Plate1 being closer to a negative contribution to EE for charge type B than was previously anticipated. Furthermore, the analysis only focused on 4 of the 12 scrap categories, out of which 3 are of particular interest to the selected charge types. In addition, the model also uses a total of 21 input variables to predict the EE consumption of any given heat. However, given these caveats, the analysis is in line with what could be expected from the model analyzed as well as from expert domain knowledge.

The main utility of SHAP main interaction values is that it provides clarity on how specific values governed by the input variable distribution contribute to the EE consumption prediction. The SHAP main interaction values only show the univariate relationship between the input variable and the EE consumption prediction. It is possible to use the SHAP interaction effects between the input variables as explained in Section 3.5.1. However, the number of SHAP plots to analyze will be equal to an additional 20 for each input variable; giving a total of 441 SHAP interaction plots for a complete analysis of the selected model. Although experience and knowledge about the specific EAF can guide the selection of SHAP interaction plots, a further analysis of the SHAP plots is best left as a future point of study. The above analysis shows what can be done with the available tools. Although interesting, an exhaustive analysis is for obvious reasons out of scope of the present paper.

The SHAP main interaction effects on EE by each scrap category of the selected model can be observed in Figure 10. Thin plate, i.e., Plate1, has been confirmed by the steel plant engineers to contribute to less EE. This is evident since a steep drop can be observed. The reduced EE by Plate1 is eventually flattened out and increases when the amount of Plate1 approaches the upper limit of the furnace capacity. Internal1 contains heavy scrap with an apparent density of over 1.4 ton/m$^3$. Heavy scrap takes longer time to melt which contributes to a steadily rising EE with an increased amount of Internal1 scrap. This has also been confirmed by the steel plant engineers, which refer to the use of Internal1 as the reciprocal of Plate1 in the steel plant charging strategies. One could observe a slight decrease in the EE contribution by increasing the amount of Plate2 and a slight increase in EE contribution for Internal2. However, these scrap categories consist of less than 1% of the total charged

weight in the studied heats. Hence, it is difficult to draw any clear conclusion on their contribution to the EE. Shredded scrap is charged based on operating practices rather than for specific charging strategies which results in steel with low amount of tramp elements. The EE contribution is decreasing with increased amount of Shredded scrap from the nominal amount used. This was not confirmed by the process engineers. The decreasing EE contribution could be a model artifact or because the shredded scrap does contribute to a decrease EE consumption in the process. The latter could be the case since shredded scrap is easily melted due to its high surface-area-to-volume ratio.

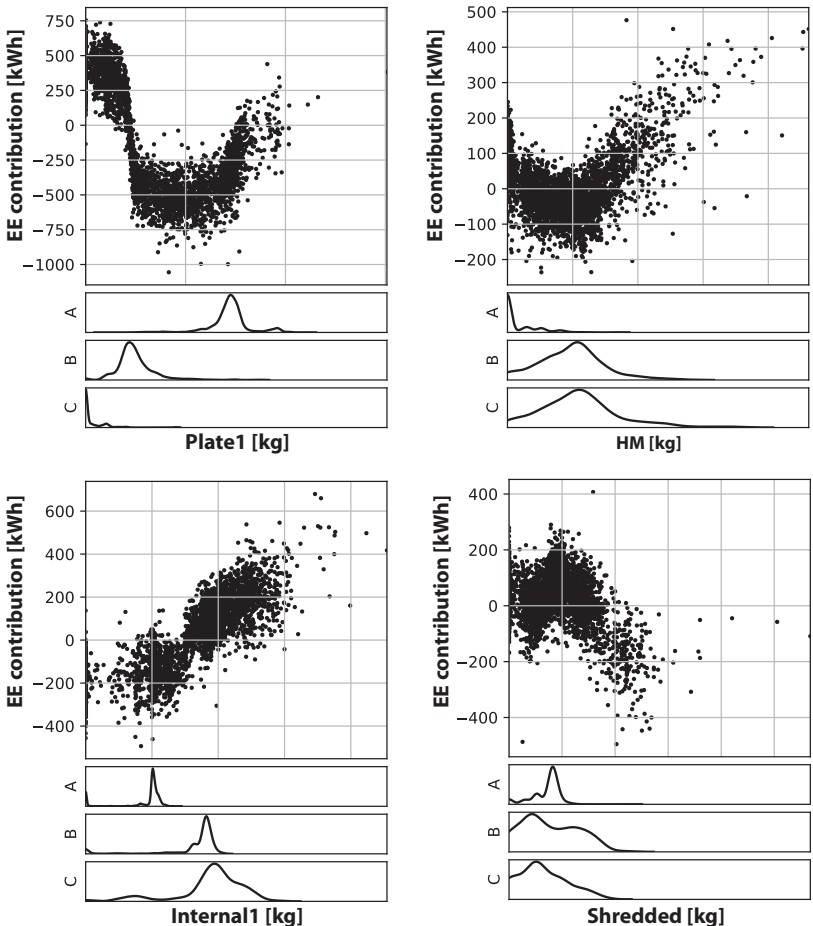

**Figure 9.** SHAP main interactions on EE by Plate1, Internal1, HM, and Shredded scrap categories. The three probability density plots below each plot show the distribution of the scrap category for each of the charge types A, B, and C, respectively. See Section 3.5.3 for an explanation of the different steel types. The y-axis of the scatter plot shows the SHAP main interactions on EE for each variable and the y-axes of the probability density plots show the frequency of each charge weight. The x-axes of both the SHAP plot and the probability density plot for each charge type are the amount charged of the scrap category. The values on the x-axes has been omitted due to proprietary reasons. The values governing the plots are from both the training and test data.

Incineration scrap is charged in low amounts, corresponding to only approximately 1.5% of the total charging weight on average for all heats and approximately 5% for the heats using charge types with lower requirements on impurity tramp elements. The increase in EE consumption is likely due to the melting requirements of the scrap weight rather than due to the surface-area-to-volume ratio. Also, skulls require higher EE according to the SHAP main interaction effect. This was confirmed by the steel plant engineers which reported that skulls are difficult to melt. Skulls are large concrete-like pieces of slag and metal mixture.

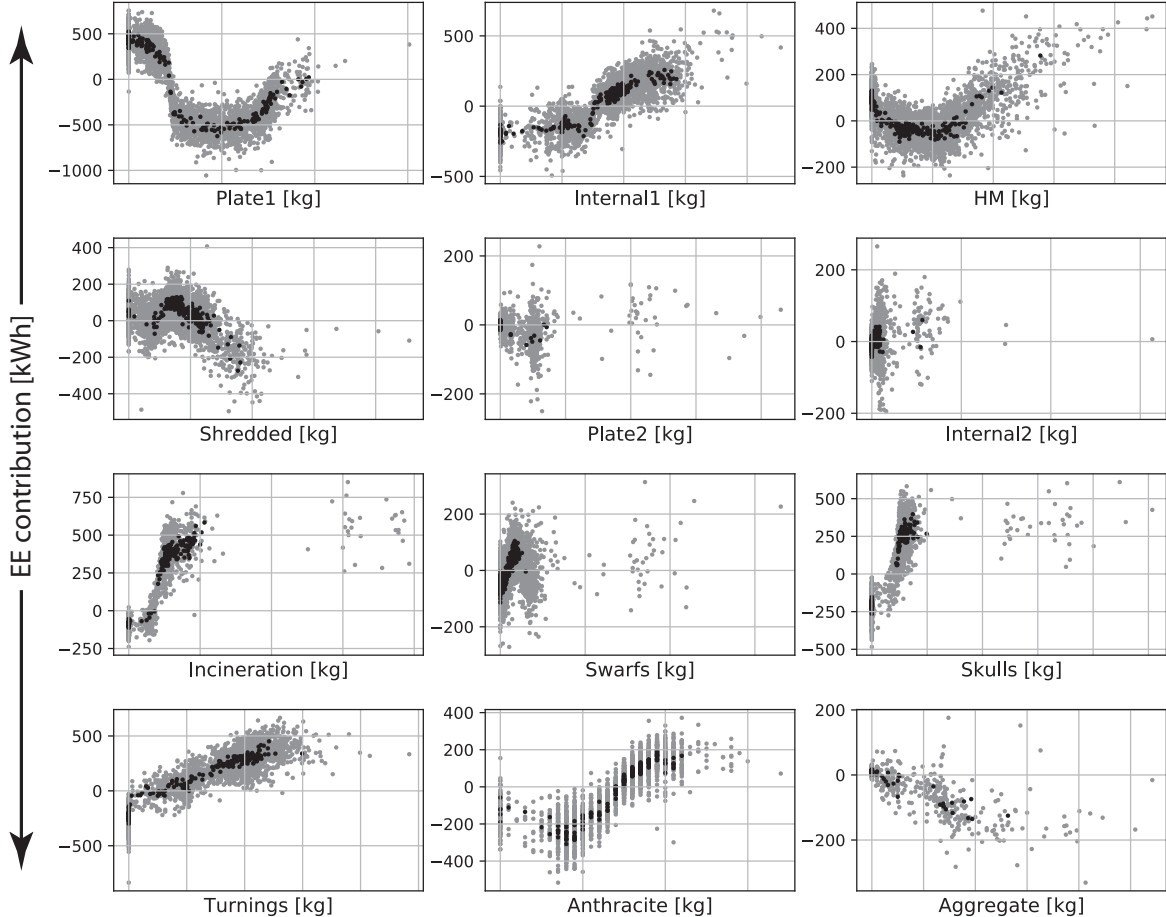

**Figure 10.** SHAP main interaction effects for the scrap categories for the selected model. The y-axes show the main interaction effect on EE while the x-axes show the amount of each charged scrap category. The values on the x-axes has been omitted due to proprietary reasons. The grey dots represent values from the training data and the black dots represent values from the test data.

The SHAP main interaction values for the base variables can be seen in Figure 11. Here, the focus will be on the variables whose EE contribution is counter-intuitive from the standpoint of a practitioner in physico-chemical modeling. Specifically, these are Burner $O_2$, Carbon Injection, Burner oil, and Lance $O_2$. According to the steel plant engineers, Burner oil is only effective up to a certain amount of $kg$, which is when the burner oxygen is used in tandem with burner oil. This agrees well with the observation from Figure 11. The burners are used in their maximum capacity when melting scrap. However, the burners still need to be active for the remainder of the heat to prevent the burners from getting clogged by slag and scrap. Thus, Burner oil is closely related to TTT, which is the reason Burner oil in higher amounts contributes positively to EE. Carbon injection, which should contribute to more heat generated by carbon boil, contributes positively to EE. This was confirmed by the steel plant engineers to be related to the continuous injection of carbon fines throughout the heat. As soon as liquid steel is present, carbon fines are injected to facilitate foaming slag. Similar to Burner oil, Carbon Injection is also closely related to TTT. In the steel plant of study, Lance $O_2$ is only used to clear the slag door to enable sampling of the steel melt temperature and composition. Therefore Lance $O_2$ does not have a consistent contribution to the EE.

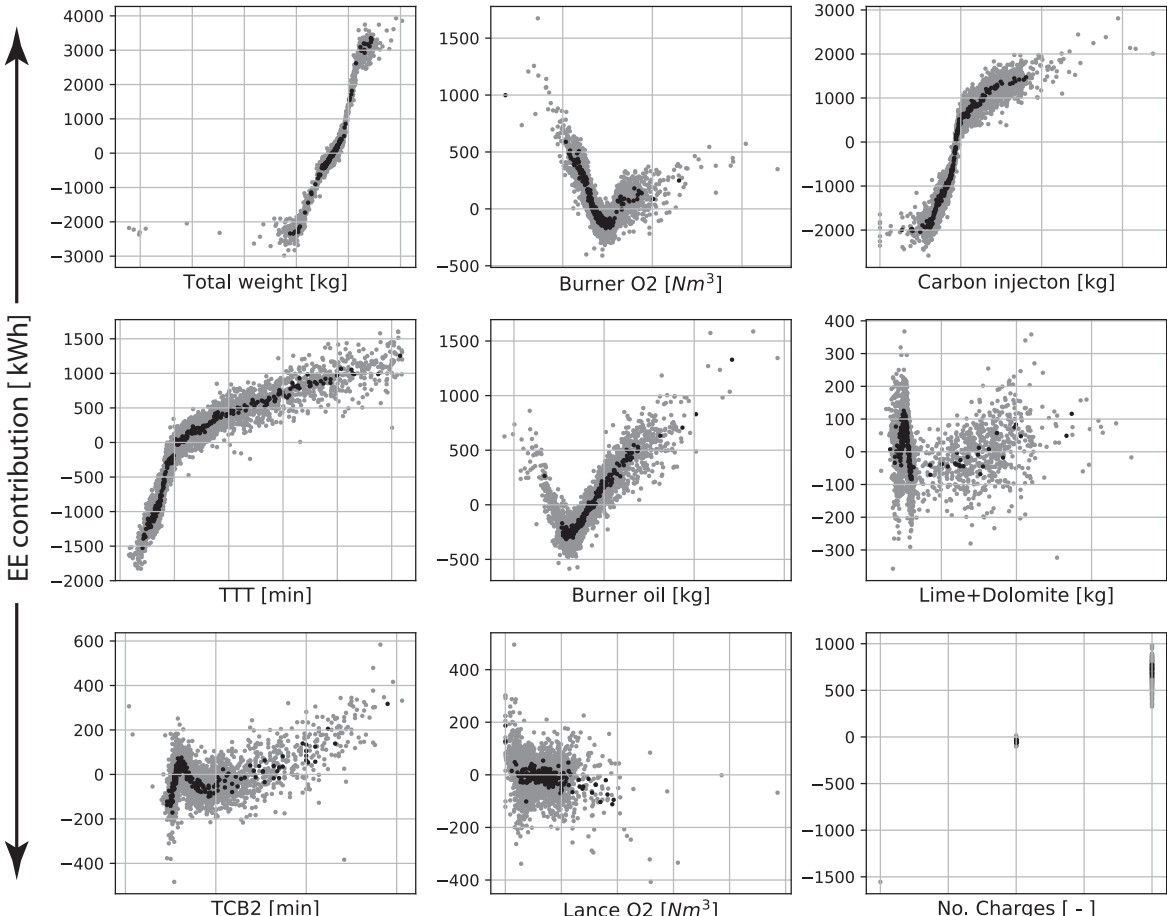

**Figure 11.** SHAP Main interaction effects for the base variables for the selected model. The y-axes show the main interaction effect on EE while the x-axes show the increasing amount of each variable. See Table 2 for details about the definition of each variable. The values on the x-axes has been omitted due to proprietary reasons. The grey dots represent values from the training data and the black dots represent values from the test data.

The positive relationship Injection Carbon to EE can also be observed by its Pearson correlation coefficient in Table 9. For both the Pearson correlation and dCor, Injection Carbon has the highest value with respect to the EE. In addition, the SHAP feature importance (Figure 12) regards the Carbon Injection as almost twice as important as the Total weight when the model predicts the EE consumption.

The sometimes counter-intuitive relations between the input variables to the EE consumption prediction emphasize the importance of not only having a firm understanding of the physico-chemical and process experience on the relations governing the EAF. It is also important to understand the relationship between the governing factors to the EE consumption of the specific EAF that one intends to model.

It is important to observe that the total weight and TTT are among the three most important variables for the model when predicting on EE. Both variables are also correctly considered by the model with respect to what is known from process metallurgical experience. This agrees well with the results from the previously reported SHAP analysis of a statistical model, created by the authors of the present study, predicting the EE of an EAF [10].

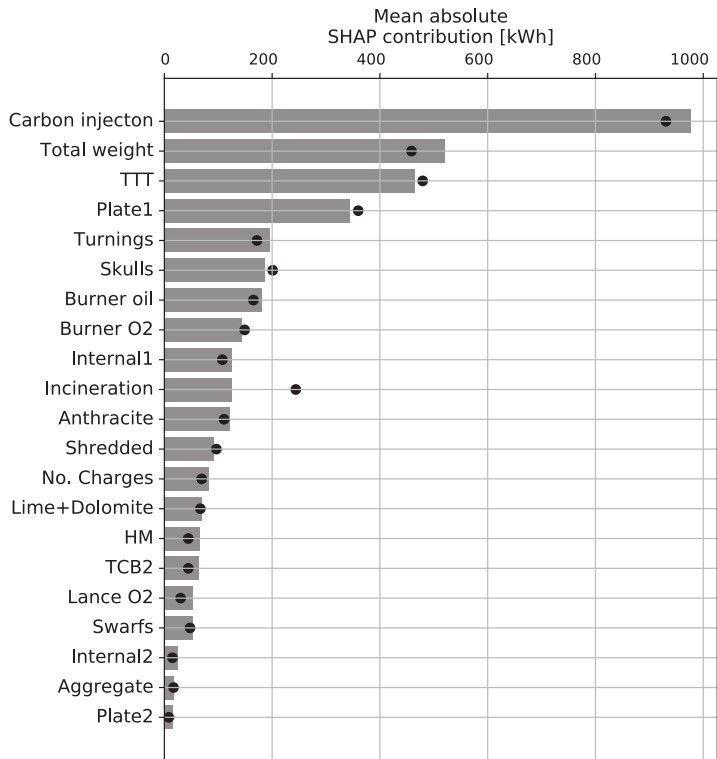

**Figure 12.** SHAP feature importance for the selected model as defined in Equation (15). The bars show the feature importance on the training data and the dark dots show the feature importance on the test data.

**Table 9.** Correlation between the input variables and the EE consumption as defined in the transformer system. The input variables are ordered by dCor in the training data set. The values in parentheses are the Pearson correlation coefficients.

| Input Variable | Training | Test | Input Variable | Training | Test |
|---|---|---|---|---|---|
| Carbon injection | 0.39(0.40) | 0.33(0.31) | TCB2 | 0.12(0.13) | 0.14(0.07) |
| Total weight | 0.38(0.36) | 0.35(0.37) | HM | 0.10(0.10) | 0.22(0.21) |
| TTT | 0.28(0.31) | 0.33(0.31) | Lance O2 | 0.09(0.07) | 0.16(0.13) |
| Burner oil | 0.24(0.24) | 0.33(0.29) | Anthracite | 0.09(0.05) | 0.23(0.17) |
| Plate1 | 0.20(-0.17) | 0.34(-0.31) | Internal2 | 0.08(0.05) | 0.34(0.06) |
| Turnings | 0.20(0.20) | 0.28(0.24) | Swarfs | 0.07(0.04) | 0.28(0.20) |
| Skulls | 0.18(0.15) | 0.35(0.34) | Shredded | 0.05(-0.03) | 0.22(0.22) |
| Charges | 0.17(0.19) | 0.20(0.21) | Lime+Dolomite | 0.04(0.01) | 0.13(-0.13) |
| Internal1 | 0.16(0.16) | 0.22(0.13) | Plate2 | 0.04(-0.02) | 0.07(-0.01) |
| Burner O2 | 0.15(0.13) | 0.14(0.0) | Aggregate | 0.04(-0.04) | 0.10(-0.07) |
| Incineration | 0.15(0.10) | 0.33(0.30) | | | |

## 5. Conclusions

The main aim of this study was to investigate the effects of scrap in the performance of a statistical model predicting the EE consumption of an EAF. This was done using three distinct representations of the scrap types as well as SHAP main interaction effects which reveals the contributions from the scrap variables on each specific EE consumption prediction. In addition, the study extended a previously reported methodology used to create statistical models predicting the EE consumption of an EAF [2]. This was achieved by investigating the effects on the model performance using four different data

cleaning strategies, an additional statistical model framework, RF, and data from an EAF producing steel for tubes, rods, and ball-bearing rings as opposed to stainless steel.

The main conclusions of the study may be summarized as follows:

### 5.1. Analysis of the Models

- All models using variables from any of the three scrap representations performed better than the models using only the total charged scrap weight as a scrap representation. This consistency provides evidence that more granular scrap representations are important when modeling the EE consumption of an EAF using steel scrap as the main raw material.
- The models using the visual scrap representation had the highest $\bar{R}^2_\mu$-values regardless of cleaning strategy. This provides evidence that a scrap representation based on the shape of the scrap, i.e., surface-area-to-volume ratio, is the most optimal to use. The result agrees well with the surface-area-to-volume ratio and the physico-chemical relationships on the scrap melting rate governed by temperature gradients, alloying gradients, and the freezing effect. In addition, this representation is also intuitive and easy to apply from the perspective of steel plant engineers and scrap yard operators.
- Data cleaning strategies using domain-specific knowledge from either an expert from the steel plant of study or a domain expert provide models with the highest $\bar{R}^2_\mu$-values compared to pure statistical cleaning methods. This further emphasizes the importance of domain-specific knowledge when modeling the EAF.

### 5.2. Analysis of the Selected Model

- The effects of the most important scrap categories on the EE consumption for the selected charge types A, B, and C are transparently presented by SHAP main interaction values.
- SHAP main interaction values revealed the contribution by each scrap category on the complete prediction space for the selected model. Plate1 and Internal1 were confirmed by the steel plant engineers to agree well with their expectations. However, a more thorough investigation must be conducted to evaluate the reasons behind the effects of the other scrap categories. SHAP interaction values between the input variables is advised as a starting point.
- The experience and knowledge provided by the steel plant engineers were essential to determine the reasons behind the counter-intuitive responses by Injected Carbon, Lance $O_2$, Burner oil, and Burner $O_2$ on the EE consumption. This indicates that the participation by steel plant engineers is of outmost importance to evaluate the trustworthiness of a practical model.
- The authors question whether one should use the Carbon Injection as part of EE prediction models in the steel plant of study, given the specific operational practice. It is likely that the Carbon Injection variable creates a model artifact for the selected model.
- The test data coincides with the training data in the SHAP plots. This raises the trust in the model since the model response on previously unseen heats is within the range of SHAP values from the heats used to adapt the model parameters.

## 6. Further Work

A continuation of the present study is to systematically evaluate the EE consumption of each scrap category by performing controlled experiments. This would require a similar approach taken to achieve the experiments in this study. Specifically, modeling according to the methodology presented in this study, categorizing the scrap according to one of the provided scrap representations, and cooperating with the steel plant engineers. As a final step, the resulting EE prediction model should be implemented in the existing EAF process control system. Only then can the true effect by the model on the resulting, actual, EE consumption be investigated.

**Author Contributions:** Conceptualization, L.S.C. and P.B.S.; methodology, L.S.C. and P.B.S.; software, L.S.C.; validation, L.S.C. and P.B.S.; formal analysis, L.S.C. and P.B.S.; investigation, L.S.C. and P.B.S.; resources, P.B.S. and P.G.J.; data curation, L.S.C.; writing–original draft preparation, L.S.C.; writing–review and editing, L.S.C. and P.B.S.; visualization, L.S.C.; supervision, P.B.S. and P.G.J.; project administration, P.B.S. and P.G.J.; funding acquisition, P.G.J. All authors have read and agreed to the published version of the manuscript.

**Funding:** This research was funded by "Hugo Carlssons Stiftelse för vetenskaplig forskning" in the form of a scholarship granted to the corresponding author.

**Acknowledgments:** We want to thank the plant engineers Patrik Undvall and Jan Pettersson at the Ovako Steel Hofors mill for their support and data provisioning during this project.

**Conflicts of Interest:** The authors declare no conflict of interest.

## Nomenclature

| | |
|---|---|
| $T_{HM}$ | Temperature of the molten steel |
| $T_{liq}$ | Scrap melting temperature |
| $\rho_{scr}$ | Density of the scrap metal |
| $H_s$ | Heat of melting of scrap |
| $c_p$ | Specific heat of scrap |
| $h$ | Heat transfer coefficient in the interface of the molten steel and scrap |
| $\beta$ | Mass transfer coefficient of carbon |
| $C_l$ | Carbon content in liquid steel |
| $C_i$ | Carbon content in the solid-liquid interface |
| $\rho_s$ | Apparent density of scrap |
| $C_0$ | Initial carbon content in the steel scrap. |
| $h_{scr}$ | Mass transfer coefficient under forced convection |
| $c$ | Experimentally determined constant |
| $p$ | Experimentally determined constant |
| $u$ | Average stirring power in the boundary between the melt and scrap surface area |
| $R_{SV}$ | Surface-area-to-volume ratio |
| $A$ | The total area of an arbitrary scrap piece |
| $V$ | The total volume of an arbitrary scrap piece |
| $l$ | The length of an arbitrary scrap piece |
| $r$ | The radius of a cylindrical or spherical scrap piece |
| $t$ | The thickness of a square plate |
| $m$ | The mass of a scrap piece |
| $T$ | Temperature of the surface of a body emitting thermal radiation |
| $q_1$ | First quartile |
| $q_3$ | Third quartile |
| $\epsilon$ | Pre-specified constant for outlier extremity |
| $x_j$ | An instance of variable $j$ |
| $\sigma$ | Standard deviation |
| $v$ | Number of input variables |
| $P$ | Number of nodes in the previous layer |
| $s_j$ | Summation of the input values for j:th node in the current layer |
| $w_i$ | Weight of node $i$ in the previous layer |
| $x_i$ | Value of node $i$ in the previous layer |
| $x_k$ | A data instance to be predicted by a RF |
| $y_k$ | The predicted value of data instance $x_k$ by a RF |
| $R^2$ | Coefficient of determination |
| $\overline{R}^2$ | Coefficient of determination adjusted for number of data points and variables |
| $n$ | Number of data points |
| $E_i$ | Regular error for data point $i$ |
| $y_i$ | True value of the output variable for data point $i$ |
| $\hat{y}_i$ | Predicted value of the output variable for data point $i$ |

| | |
|---|---|
| $\overline{R}^2_\mu$ | Mean adjusted R-square of the 10 model instances on test data |
| $\overline{R}^2_\sigma$ | Standard deviation of adjusted R-square of the 10 model instances on test data |
| $\overline{R}^2_{\min}$ | Minimum of adjusted R-square of the 10 model instances on test data |
| $\overline{R}^2_{\max}$ | Maximum of adjusted R-square of the 10 model instances on test data |
| $\Delta_\mu$ | Mean error of the mean error of the 10 model instances on the test data |
| $\Delta_\sigma$ | Standard deviation of the mean error of the 10 model instances on the test data |
| $\Delta_{min}$ | Minimum of the mean error of the 10 model instances on the test data |
| $\Delta_{max}$ | Maximum of the mean error of the 10 model instances on the test data |
| $f$ | An arbitrary statistical model |
| $f_x$ | A simplified representation of $f$ |
| $\phi_0$ | The contribution to the prediction by $f$ when all information is absent |
| $\phi_i$ | The contribution to the prediction by $f$ by variable $i$ |
| $Z$ | A subset of a set of all input variables |
| $\hat{Z}$ | The set of all input variables |
| $|Z|$ | The number of variables in the subset $Z$ |
| $\hat{Z} \setminus \{i\}$ | The set of all input variables excluding variable $i$ |
| $Z \cup \{i\}$ | The subset of a set of all input variables including variable $i$ |
| $Z \cup \{j\}$ | The subset of a set of all input variables including variable $j$ |
| $Z \cup \{(i,j)\}$ | The subset of a set of all input variables including variable $i$ and $j$ |
| $M$ | The total number of input variables |
| $\phi_{i,i}$ | The main interaction value for variable $i$ |
| $\phi_{j,i}$ | The interaction effect of input variable $i$ imposed on variable $j$ |
| $\phi_{i,j}$ | The interaction effect of input variable $j$ imposed on variable $i$ |
| $FI_j$ | SHAP feature importance for variable $j$ |
| $\phi_j^i$ | Regular SHAP value for variable $j$ and data instance $i$ |
| $\rho_{X,Y}$ | Pearson correlation coefficient between random variables X and Y |
| $\sigma_X$ | Standard deviation of random variable X |
| $\sigma_Y$ | Standard deviation of random variable Y |
| $V_1$ | Random variable |
| $V_2$ | Random variable |
| $dCor(V_1, V_2)$ | Distance correlation between $V_1$ and $V_2$ |
| $dCov(V_1, V_2)$ | Distance covariance between $V_1$ and $V_2$ |
| $\sqrt{dVar(V_1)}$ | Distance standard deviation for $V_1$ |
| $\sqrt{dVar(V_2)}$ | Distance standard deviation for $V_2$ |

## Abbreviations

The following abbreviations are used in this manuscript:

| | |
|---|---|
| EE | Electrical Energy |
| EAF | Electric Arc Furnace |
| BOF | Basic Oxygen Furnace |
| DRI | Direct Reduced Iron |
| DMS | Demand Side Management |
| MLR | Multivariate Linear Regression |
| PLS | Partial Least Squares regression |
| ANN | Artificial Neural Network |
| RF | Random Forest |
| KS | Kolmogorov–Smirnov |
| dCor | Distance Correlation |
| SHAP | SHapley Additive Explanations |
| TTT | Tap-to-Tap Time |
| TCB2 | Time to Charging of Basket 2 |
| HM | Heavy Melting Scrap |
| VB | Variable Batch |

# Appendix A

**Table A1.** Hardware specifications for the numerical experiments.

| | |
|---|---|
| Computer model | Dell Latitude E5570 |
| CPU | Intel Core i7 2376 MHz |
| RAM | 16,203 MB |

**Table A2.** Software specifications for the numerical experiments.

| Purpose | Software/Package | Version |
|---|---|---|
| Operating system | Microsoft Windows 7 Professional | 6.1.7601 Service Pack 1 Build 7601 |
| Programming language | Python 3 | 3.7.1 |
| Python distribution | Anaconda 3 | 4.6.7 |
| Data handling | pandas | 0.23.4 |
| | numpy | 1.17.4 |
| Statistical modeling | scikit-learn | 0.20.1 |
| SHAP values | shap | 0.8.1 |
| Distance correlation | dcor | 0.3 |
| Visualization | matplotlib | 3.0.2 |
| | seaborn | 0.9.0 |

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
