# Peer review of "Modeling the Effect of Scrap on the Electrical Energy Consumption of an Electric Arc Furnace"

_processes, doi:10.3390/pr8091044_

Round 1

Reviewer 1 Report

The authors present an interesting study on the energy consumption of the EAF process which is highly relevant for waste management, as often LCA is applied to evaluate the environmental impact of recycling processes, but the availability of appropriate input data such as the energy consumption represents a significant challenge.

The article is written very well and gives great insights into the processes of steelmaking and modelling.

I have the following detailed remarks for further improvement of the article:

line 18: In the sentence before you talk about the energy, now you switch to the energy losses. This is unclear. It should be pointed out how many percent of the used electrical energy are used for the melting and how much is lost, i.e. the efficiency of the process should be quantified.

line 44: Scrap is classified according to the European Scrap Type listing. Thus, it would be beneficial for EAF plants to link the Scrap Type with the parameters of your study. Could you do this or argue why it is not possible?

Equation 1: In line 66 you mention the melting rate m/s, but in the equation there is dx/dt. I think that x is m/s, but this should be mentioned.

General remark: Also in an EAF plant, slag forming agents, e.g. limestone, are added. Their effect on the melting of scrap should be considered as well!

Table 1: This is a very good overview. However, it shows enormous differences between the different studies. Thus, it would be great to include some sentences on the reasons of this, such as "Miller et al provide the lowest value for ... which is because in their plant high-carbon steel is produced..." etc. so that the reader gets an overview which factors influence the distribution of energy between the compartments.

Author Response

// Thank you for your valuable feedback and suggestions. Specific answers to each part of your review can be viewed after “//” in each section, in bold text.

Any text changes in the manuscript has been highlighted using the \highlight command in LaTeX, which will be visible as yellow coating over the affected text snippet. Additional information, if needed, is presented in the answer to the corresponding feedback/suggestion provided by you.

Reviewer 2 Report

Reviewer’s comments on manuscript Processes-899423

The submitted paper presents the outcomes of statistical modeling of charge scrap type on electricity consumption in a scrap melting furnace. Underlying data were provided from a real steel plant and the model construction, modeling procedure and its outcomes were consulted with company’s experts. Such direct application-oriented approach is highly appreciable.

A few recommendations, listed below, should be considered in a minor revision before this manuscript is accepted for publication. Apart from this, the manuscript is well-prepared, scientifically sound and falls within the scope of Processes journal.

  1. Abstract is of adequate length and provides concise information about the manuscript goals and achievements. However, it lacks any quantitative information which would support the conclusions drawn in abstract. Though I understand it is not straightforward to draw any quantitative conclusions from statistical modeling, the authors should endeavor to enrich the abstract by this aspect.
  2. Authors stress the practical usefulness of the model and that is should be easily and intuitively used by plant engineers. It would further increase the manuscript significance and value, if additional information about this aspect is provided: Can you describe and perhaps quantify the benefits it brought to the steel plant engineers? Maybe in terms of charge preparation process simplification, its shortening, improved production management, shortened charge processing time, or lowered electricity/fuel consumption, or other benefits? If you are able to do so, please revise the manuscript significance statement (lines 36 to 50) accordingly.
  3. Equation (3) includes the stirring speed. This is a vague term. Please explain what exactly is understood by it.
  4. Introduction + model background and modeling procedure takes up around 700 lines, whereas the results take around 140 lines. I think the manuscript is imbalanced in this sense. I recommend condensing the modeling part and shift the less important parts to appendices. My candidate for this is, for example, the explanation of the physico-chemical background of the problem.
  5. Please revise the Nomenclature section and include symbols, abbreviations and variables used in text but not mentioned there.
  6. Please revise and unify the formatting style in References section according to the journal’s guide for authors. The headline “References” is missing.

Author Response

(The authors gave the same response as above.)

Reviewer 3 Report

In the paper a modeling the electrical energy consumption based on shape of the scrap is presented.

Major concerns

  • The authors published several similar “EAF energy consumption modeling” papers but still “shape of the scrap” influences seem optimistic…
  • Due to main theme “effects of shape/geometry of scrap on electric energy consumption” the introduction should involve at least a few shortly presented findings/papers related to “scrap geometry”…
  • At least the effects of other input variables can be briefly presented in the introduction section (studies 2, 10)
  • Could the actual delays in subsection 2.2.4 be evaluated, presented (e.g. percent, table, graph)?
  • The conclusion could contain also future work presentation. It is already mentioned that the effects of scrap categories should be elaborated… How could be practical implementation conducted?

Minor concerns

  • The average annual consumption of individual scrap category could be presented (e.g. graph, table)
  • The typical production steel chemical composition could be presented (min., max. content)
  • Spaces before units (e.g. “100kg”, “10%”)
  • Axes on figures 9 and 11 could be equipped with the units (e.g. kWh, kg)
  • Could the surface-to-volume ratios be added in subsection 3.1.2?
  • The test data from February 2020 was used… Till when? How many batches? For test and training data?

Author Response

(The authors gave the same response as above.)

Reviewer 4 Report

A very comprehensive document which details results of an investigation into the modelling of scrap melting on electrical energy consumption. This is a topic of current interest and the authors have produced a very impressive manuscript, which demonstrates novelty, and adds to the body of knowledge in this field, as well as being of interest to industry. The paper is long, but I feel that the length is just about justified, as the vast majority of the information presented is directly relevant to the topic. In particular the background information in section 2, and the descriptions of the various steps employed in the model selection and training of the test data are useful for the reader in understanding the evolution of the methodology. There are only a few sections that could be edited for length - maybe some of the information in section 3.4.1 for example could have been removed without decreasing the impact. The results and discussion section is well presented, and the conclusions and analysis are supported by the experimental observations. The quality of presentation is generally high, and the figures are clear. There are many grammatical errors, but these are relatively minor and can be picked up during the proofreading process. I recommend acceptance with minor revision - mainly to improve the English

Author Response

(The authors gave the same response as above.)

Round 2

Reviewer 3 Report

I believe that all issues have been resolved… consequently, I recommend this paper for publication.

Author Response

Thank you!